



# Towards Effective Drought Monitoring in the Middle East and North Africa (MENA) Region: Implications from Assimilating Leaf Area Index and Soil Moisture into the Noah-MP Land Surface Model for Morocco

Wanshu Nie[1,2], Sujay V. Kumar[3], Kristi R. Arsenault[3,4], Christa D. Peters-Lidard[5], Iliana E. Mladenova[6], Karim Bergaoui[7], Abheera Hazra[3,8], Benjamin F. Zaitchik[1], Sarith P. Mahanama[9,10], Rachael McDonnell[11], David M. Mocko[3,4], Mahdi Navari[3,8]

[1] Department of Earth and Planetary Sciences, Johns Hopkins University, Baltimore, Maryland, USA
[2] NASA Goddard Earth and Sciences Technology and Research (GESTAR), Greenbelt, Maryland, USA
[3] Hydrological Science Laboratory, NASA Goddard Space Flight Center, Greenbelt, Maryland, USA
[4] Science Applications International Corporation, McLean, Virginia, USA
[5] Earth Science Division, NASA Goddard Space Flight Center, Greenbelt, Maryland, USA
[6] USDA Foreign Agricultural Service, Washington, DC, USA
[7] ACQUATEC Solutions, Dubai Technology Entrepreneur Campus (DTEC), Dubai Silicon Oasis, Dubai, UAE
[8] Earth System Science Interdisciplinary Center, University of Maryland, College Park, Maryland, USA
[9] Science Systems and Applications Inc., Lanham, Maryland, USA
[10] Global Modeling and Assimilation Office, NASA Goddard Space Flight Center, Greenbelt, Maryland, USA
[11] Water, Climate Change & Resilience Program, International Water Management Institute-Rome Office, Rome, Italy

*Correspondence to:* Wanshu Nie (nwanshu1@jhu.edu)

**Abstract.** The Middle East and North Africa (MENA) region has experienced more frequent and severe drought events in recent decades, leading to increasingly pressing concerns over already strained food and water security. An effective drought monitoring and early warning system is thus critical to support risk mitigation and management by countries in the region. Here we investigate the potential for assimilation of leaf area index (LAI) and soil moisture observations to improve 25 representation of the overall hydrological and carbon cycles and drought by an advanced land surface model. The results reveal that assimilating soil moisture does not meaningfully improve model representation of the hydrological and biospheric processes for this region, but rather it degrades simulation of interannual variation of evapotranspiration (ET) and carbon fluxes, mainly due to model weaknesses in representing dynamic phenology. However, assimilating LAI leads to greater improvement, especially for transpiration and carbon fluxes, by constraining the timing of simulated vegetation growth 30 response to evolving climate conditions. LAI assimilation also helps to correct for the erroneous interaction between the dynamic phenology and irrigation during summertime, effectively reducing a large positive bias in ET and carbon fluxes. Independently assimilating LAI or soil moisture alters the categorization of drought, with the differences being greater for more severe drought categories. We highlight the vegetation representation in response to changing land use and hydroclimate as one of the key processes to be captured for building a successful drought early warning system for the 35 MENA region.



# 1 Introduction

The Middle East and North Africa (MENA) region has experienced intensified drought events during recent decades that are attributable to climate change (Bergaoui et al., 2015; Cook et al., 2018; Pachauri et al., 2014). Multiple dimensions of food, water, and energy security are affected by drought in the MENA countries, leading in some instances to increased social disparities, political disruption, and disease outbreaks (Müller et al., 2017; Rajsekhar and Gorelick, 2017; Stanke et al., 2013; Weinthal et al., 2015). While efforts have been made to cope with these extreme events through the engagement of multi-sectoral and interdisciplinary collaborations among communities and across scales, strategic drought risk management linked to operational drought early warning systems for the region are not yet in place (Pulwarty and Sivakumar, 2014; Verner et al., 2018). Recognizing the complexity in defining drought and the broad range of drought impacts, drought experts have favored using a Composite Drought Indicator (CDI) approach that combines indicators from different climate variables such as precipitation, soil moisture, vegetation and evapotranspiration (ET) into a single product through a "convergence of evidence" framework (Hayes et al., 2012). Such approaches have been applied to four MENA countries (i.e., Morocco, Tunisia, Lebanon and Jordan) through a USAID-funded project in developing the MENA Regional Drought Management Systems (MENA RDMS; Jedd et al., 2020; Bijaber et al., 2018; Fragaszy et al., 2020). Accurate estimates of these constituent variables are therefore necessary for developing reliable drought assessments with CDI. Advances in remote sensing and earth observation technologies offer the ability to infer precipitation, soil moisture, and vegetation from different sensing platforms at various spatiotemporal resolutions. While these measurements offer valuable information on the changes in land surface conditions, they suffer from spatio-temporal gaps in coverage from orbital configurations and sensing limitations, and, for soil moisture, a limitation to shallow sensing depths (~ a few centimeters), restricting their direct use for drought representation (Brocca et al., 2017; Kerr et al., 2016). Extending the utility of spatially incomplete and temporally infrequent remote sensing data to improve the representation of variables such as root zone soil moisture is needed for improving CDI estimation.

Earth system models are powerful tools to generate continuous soil moisture profiles with full spatiotemporal coverage and provide additional information on the distribution of water resources, carbon fluxes, and their impact on hydroclimate over a broad range of scales. However, because of uncertainties in simplified model physics in representing complex real-world systems and deficiencies inherent in meteorological forcing inputs, soil and vegetation parameters, model-based estimates are subject to error. Ground measurements can serve to constrain the estimation of model parameters, but such observations are very limited or unavailable for the MENA region. Therefore, one approach to mitigate the model uncertainties and extend the value of remote sensing products is to merge them through data assimilation (DA; Reichle, 2008).

Assimilating remote sensing measurements that contain information on either surface soil moisture or vegetation can affect the simulation of root zone soil moisture. Evidence from previous soil moisture DA studies suggests that DA can provide





higher skill in estimating both surface and root zone soil moisture, benefiting applications such as agricultural drought
monitoring and irrigation management (Bolten et al., 2009; De Lannoy and Reichle, 2016; Kolassa et al., 2017a; Kumar et
al., 2012; Lei et al., 2020; Liu et al., 2011). In particular, the improvement could be larger in data-sparse regions as the
model skill without assimilation in data-rich regions such as in the U.S. is generally high owing to abundant ground
reference datasets (Kolassa et al., 2017a). Similarly, vegetation related products, such as leaf area index (LAI) can also
inform the variations in surface and root zone soil moisture by influencing water uptake and the partitioning of
evapotranspiration. Prior studies have shown that assimilating LAI helps to improve the estimation of evapotranspiration,
root zone soil moisture, carbon fluxes and crop yields (Albergel et al., 2017; Barbu et al., 2014; Ines et al., 2013; Kumar et
al., 2019b; Mocko et al., 2021; Xie et al., 2017).

In this study, taking the northern part of Morocco as an example, we report on the separate assimilation of soil moisture and
LAI into the Noah-MP land surface model (LSM), in a configuration where the input parameters and meteorological forcing
data sets are customized for the MENA region. In this region, much of the agricultural lands are dominated by rain-fed
agriculture. However, irrigation activities may still play a critical role in altering the root zone soil moisture and the
associated energy and carbon fluxes for irrigated areas. The water demand for irrigation may also vary widely in a changing
climate that may include persistent droughts (Kharrou et al., 2011). Therefore, we set up the LSM simulations both with and
without the presence of such human water management. Our objectives are twofold: first, we investigate the overall
performance of the data assimilation system by comparing the simulated ET and carbon fluxes with a variety of multi-source
remote sensing datasets; second, we examine the potential of data assimilation to improve representation of root zone soil
moisture - based drought, which serves as an input for the CDI estimates for the MENA region (Bijaber et al., 2018).

## 2. Materials and Methods

### 2.1 Model configuration

All the simulations are conducted using the Noah-MP LSM, Version 4.0.1, implemented within the framework of the NASA
Land Information System (Kumar et al., 2006) (open source software available at https://github.com/NASA-LIS/LISF/).
Building on the Noah LSM (Ek et al., 2003), the Noah-MP model advances its structure by including multi-physics options
for radiation transfer, dynamic phenology, surface water infiltration, runoff, and groundwater schemes. A detailed
description of the model and its performance can be found in Niu et al. (2011) and Yang et al. (2011).

Noah-MP is configured with four soil layers with the layer thicknesses varying from 0.1, 0.3, 0.6 and 1 m from the surface
down to the bottom, making a total of 2 m. Water movement in the soil layers is simulated using the Richards equation. A
simple groundwater reservoir beneath the soil layer allows for soil moisture - groundwater interaction and related runoff
production. Noah-MP allows for the prognostic representation of vegetation growth using a dynamic phenology scheme
(Dickinson et al., 1998), in combination with a Ball-Berry photosynthesis-based stomatal resistance scheme (Ball et al.,
1987; Bonan, 1996; Collatz et al., 1991). It simulates the carbon uptake and allocation among leaf, stem, wood and root in



response to cold and drought stress, thus inferring the seasonal growth of leaf area and predicting carbon fluxes, such as gross primary production (GPP) and net primary production (NPP). The LAI is calculated from leaf carbon mass by multiplying by the specific leaf area. The greenness vegetation fraction (GVF), which divides a grid cell into a fractional vegetated area and a fractional bare ground area, is derived from LAI based on a simple exponential function. On one hand, vegetation photosynthesis rate is partially constrained by water stress through leaf assimilation, which is a function of the

soil moisture controlling factor. On the other hand, the canopy growth condition can in turn affect moisture partitioning via GVF, thus affecting the partition of water and energy fluxes, such as evaporation and transpiration. Given the way Noah-MP represents vegetation phenology, assimilating either surface soil moisture or LAI can alter the soil moisture and vegetation conditions, as well as the partitioning of water, energy, and carbon fluxes.

### 2.2 Data assimilation configuration

#### 2.2.1 SMAP soil moisture data assimilation (SSM-DA)

This study makes use of the SMAP Enhanced Level 3 (L3_E) Passive soil moisture estimates derived by the National Aeronautics and Space Administration Jet Propulsion Laboratory (O'Neill et al., 2020). This SMAP L3_E product provides gridded soil moisture retrievals at 6:00 am (descending) and 6:00 pm (ascending) local time on a 9-km Earth-fixed grid along with the ancillary data and quality assessment flags, starting from 31 March 2015. The assimilation is performed using

a one-dimensional ensemble Kalman filter (EnKF; Reichle et al., 2002), similar to the assimilation strategy employed in Kumar et al. (2014), which allows the update and propagation of a selected set of model states on the basis of relative uncertainty between the observations and the model ensemble. To address the biases between the SMAP retrievals and Noah-MP soil moisture, the SMAP soil moisture estimates are rescaled to the model climatology at a monthly scale for the period of 2015-2019 using cumulative density function (CDF) matching (Reichle and Koster, 2004). An ensemble of 20

members is generated by perturbing meteorological forcing fields and the surface soil moisture state, representing the uncertainty of the model estimates. The temporal correlation of the perturbation was chosen to be 1 day for the forcing fields and 1 h for the surface soil moisture state via a first-order autoregressive model. The observation error standard deviation for the unscaled soil moisture retrievals is set to 0.04 m3/m3 with a temporal correlation of 1 day, following Kumar et al. (2019a). Moreover, quality control flags were imposed so that we assimilate only data points recommended by the SMAP

retrieval quality flag for unfrozen soils with vegetation water content less than 5 kg/m2. Perturbation bias corrections following Ryu et al. (2009) were applied to all the perturbed forcing fields and surface soil moisture state to avoid biases introduced by nonlinear process in the model. A summary of the perturbation settings is shown in Table 1.

#### 2.2.2 MODIS LAI data assimilation (LAI-DA)

The level 4, 8-day composite LAI product with 500 m pixel size is used for LAI data assimilation, obtained from the

MCD15A2H Version 6 Moderate Resolution Imaging Spectroradiometer (MODIS) product, starting from July 2002





(Myneni et al., 2015). The algorithm chooses the best pixel available from all the acquisitions of MODIS sensors located on both NASA's Terra and Aqua satellites. Similar to the DA configuration for SSM-DA, an ensemble size of 20 members with the same meteorological forcing perturbation settings are set for LAI-DA using the EnKF algorithm. The model state vector in this DA instance only includes the LAI variable. Once LAI is updated from the assimilation, the leaf carbon mass gets
updated by dividing LAI with the specific leaf area. Additive perturbation with a standard deviation of 0.01 (-) is applied for both modeled and observed LAI fields with a temporal correlation of 1 h. Two layers of quality control flags are applied to select reliable retrievals prior to the assimilation in order to ensure retrieval quality (MODIS15, 2020): 1) FparLai quality flag MODLAND_QC =0 (good quality main algorithm with or without saturation), and 2) SFC_QC flag 000 or 001 (those only used the main (RT) algorithm). As cloud gaps often lead to spatial discontinuities in the MODIS-based LAI
observations, an 8-day climatological dataset is used for gap-filling in such instances (Kandasamy et al., 2013). The climatological data is generated using the entire record of LAI retrievals during the period of July 2002 – July 2020. The same quality control flags are also applied while generating this climatological dataset. Daily LAI observations are generated by linearly interpolating between the 8-day values and assimilation is then conducted at a daily time step.

## 2.3 Irrigation

In order to account for agricultural water use activities and their impact on hydrological and carbon fluxes, we utilize a demand-driven sprinkler irrigation scheme, which was introduced into Noah-MP in Nie et al. (2018), building upon the work of Ozdogan et al. (2010). The irrigation scheme works according to three key rules, including 1) where to irrigate; 2) when to irrigate; and 3) how much to irrigate.

The model's irrigated areas are identified using a composite irrigation fraction map. This irrigation map was generated by
combining the following three irrigation datasets: Global Rain-fed, Irrigated, and Paddy Croplands (GRIPC; Salmon et al., 2015), Global Irrigated Areas (GIA; Meier et al., 2018), and the International Water Management Institute's Global Irrigated Area Mapping (GIAM; Thenkabail et al., 2009) product. The 500 m GRIPC, 1 km GIA, and the 250 m GIAM products were each aggregated and converted into irrigated areal percentage at the 0.05° resolution grid of the study domain, using the Land surface Data Toolkit (LDT; Arsenault et al., 2018). For each 0.05° gridcell, the following irrigation criteria were
applied where at least two of the irrigation products had at least 1% minimum irrigated areal percentage present prior to being averaged into the composited irrigation map. The final composite irrigation fraction intensity map is shown in Figure 1(b).

The timing of irrigation — the growing season — is determined as the time period when GVF is greater than a certain threshold. In the original Ozdogan et al. [2010] implementation, the irrigation onset is specified based on a prescribed GVF
monthly climatology dataset. However, using a prescribed GVF profile is no longer suitable when the dynamic phenology module is enabled, as this may introduce inconsistency between the prescribed GVF dataset and the dynamic phenology-informed GVF. In this study, we modified the modeling system by passing the dynamic phenology module simulated GVF to the irrigation scheme, so that the growing season (during which irrigation occurs) is informed by the prognostic vegetation





conditions, which could be impacted by both soil moisture and LAI data assimilation. This is critical for places with
intensely irrigated agriculture as the interaction between the irrigation scheme, the dynamic phenology scheme, and data
assimilation enables the model to simulate the interannual variability of irrigation water use, which can directly affect the
soil moisture and indirectly affect vegetation growth through the water stress factor. Once irrigation is triggered, the
irrigation water amount is calculated as the volume required to bring root zone soil moisture deficit up to field capacity,
allowing vegetation to operate without transpiration stress. The depth of the effective root zone used for calculating irrigation
water requirement varies with time, which is a function of GVF and a crop-type-dependent parameter – maximum root
depth. Note that this study does not account for the irrigation source water partitioning, due to limited in situ observational
data for model calibration. This might affect the simulation of the surface and groundwater storage variations and deep soil –
groundwater interaction, such as capturing groundwater depletion in major aquifers due to excessive groundwater use
(Hssaisoune et al., 2020). The influence and impact on the deeper groundwater components are ignored here, as the current
study is focused on improving the soil moisture-based drought estimation.

### 2.4 Experiment design

In order to study the impacts of SSM-DA and LAI-DA in simulating the modelled water, energy, and carbon fluxes and the
impacts of irrigation on the data assimilation performance, two sets of experiments, each including an open loop (OL) run
and two data assimilation runs (with and without irrigation), are performed over northern part of Morocco (Figure 1(a)). The
model runs are conducted at 0.05° spatial resolution with a 15-minute time step, and the duration of the integrations varies
depending on the availability of the observations to be assimilated. A 57-year spin-up simulation was performed (three times
over the period of 2000-2019) to provide the initial conditions for the two sets of experiments.

Open loop experiments (OL and OL$_{irr}$): The model is run for the period 2000-2019 without assimilating any observations.
Irrigation is turned off for OL and on for OL$_{irr}$, with the prognostic GVF simulated by the dynamic phenology module
informing the timing for irrigation in OL$_{irr}$. Several sensitivity tests are performed to determine the best set of irrigation
parameters to capture the general growing season for Morocco according to the report of Global Information and Early
Warning System on Food and Agriculture (GIEWS) from the Food and Agricultural Organization of the United Nations
(FAO). The OL represents the baseline of the model skill customized for the study domain against any potential skill
improvements from assimilating either SMAP soil moisture estimates or MODIS LAI retrievals. Comparison between OL
and OL$_{irr}$ enables us to investigate the impact of irrigation when interacting with the dynamic phenology module.

SMAP-based soil moisture assimilation (SSM-DA and SSM-DA$_{irr}$): In the SSM-DA (SSM-DA$_{irr}$) experiment, SMAP L3_E
soil moisture estimates are rescaled based on the climatology of OL (OL$_{irr}$) and are then assimilated into Noah-MP for the
time period of Mar 2015 – Dec 2019. Irrigation is turned off for SMAP-DA and on for SMAP-DA$_{irr}$. Assimilating soil
moisture may affect the irrigation frequency and amount by altering the surface and root zone soil moisture condition, as it
may change the timing when the threshold of root zone soil moisture condition is reached, which serves as a check to
determine whether this area is dry enough to be irrigated.



MODIS-based LAI assimilation (LAI-DA and LAI-DA$_{irr}$): In the LAI-DA and LAI-DA$_{irr}$ experiments, the gap-filled and interpolated LAI retrievals are assimilated into Noah-MP during the time period of 2002-2019. Irrigation is turned off for LAI-DA and on for LAI-DA$_{irr}$. LAI-DA$_{irr}$ may affect irrigation in a different way as compared to SSM-DA$_{irr}$ as the observed

LAI may alter the magnitude and phase of the vegetation conditions. The changes in LAI affect the evolution of GVF, which serves to determine the growing season during which irrigation occurs. It may also indirectly affect irrigation frequency and magnitude by influencing the root zone soil moisture, which in turn alters the level of transpiration under different vegetation growth conditions.

All simulations are forced by the combination of two surface meteorology datasets, with the Integrated Muti-satellitE

Retrievals for Global Precipitation Measurement (IMERG; Huffman et al., 2015) near real-time early run providing precipitation and the National Oceanic and Atmospheric Administration (NOAA)'s Global Data Assimilation System (GDAS; Derber et al., 1991) providing the remaining set of meteorological fields, including 2-m air temperature, 2-m specific humidity, 10-m wind speed, surface pressure, and incoming shortwave and longwave radiation. Both lapse rate and slope/aspect-based topographical corrections are applied to the input meteorology to represent topographic influences on

temperature, humidity, pressure, and radiation. The model parameters include the Moderate Resolution Imaging Spectroradiometer-International Geosphere Biosphere Program (MODIS-IGBP; Friedl et al., 2010) land cover data set (1 km); the machine learning based 250-m soil property and class data set generated at the International Soil Reference Information Centre (ISRIC; Hengl et al., 2017); and the Shuttle Radar Topography Mission elevation at 30 m (Farr et al., 2007). All parameter datasets are resampled to the model resolution of 0.05o using LDT, as noted above.

The assessment for these experiments is organized as follows to serve the scientific objectives for this study: (1) we compare the differences in modelling skill among OL, SSM-DA and LAI-DA by evaluating the fluxes with available reference datasets within the overlapping period (2015-2019); (2) we then exclude SSM-DA and extend the evaluation for OL and LAI-DA for a longer time span (2003-2019); (3) we investigate the irrigation impact and its interaction with the dynamic phenology informed by data assimilation by comparing OL, OL$_{irr}$, SSM-DA$_{irr}$, and LAI-DA$_{irr}$; and (4) we quantify the

differences in categorizing drought among OL, SSM-DA and LAI-DA.

## 2.5 Evaluation data and metrics

Multi-source remote-sensing based observations covering different periods of time are used to assess the overall model performance, including ET and its components, NPP, GPP, and sun-induced chlorophyll fluorescence (SIF).

### 2.5.1 FAO WaPOR data sets

The FAO portal to monitor Water Productivity through Open access of Remotely sensed derived data (WaPOR) provides estimates of evapotranspiration (ET) and its components including bare soil evaporation (E), transpiration (T) and interception for Africa and the Middle East from 2009 onwards at dekadal time steps at three different spatial resolutions (250 m, 100 m and 30 m) with different spatial coverage. It also provides yield related variables such as net primary





production and total biomass production. WaPOR estimates E and T using a modified version of the Penman-Monteith

equation (FAO, 2020), with input from weather, land cover, NDVI, and soil moisture stress from other sources. The weather
data is obtained from the Modern-Era Retrospective analysis for Research and Applications (MERRA) up to the start of 21
February 2014 and the Goddard Earth Observing System (GEOS-5) after 21 February 2014 (Rienecker et al., 2011) in
combination with the Climate Hazards Group Infrared Precipitation with Stations (CHIRPS; Funk et al., 2015). Unlike the
original ET-Look Model (Bastiaanssen et al., 2012), in which soil moisture stress is derived from passive microwave data,

WaPOR advances the estimation of soil moisture stress by using MODIS-based land surface temperature. For NPP
estimation, besides the input for ET, the fraction of photosynthetically absorbed radiation by green vegetation (fAPAR) is
also needed, which is obtained from MODIS. Note that effects such as nutrient deficiencies, pests and plant diseases are not
considered in the calculation of the WaPOR NPP, which is also true for Noah-MP simulated NPP. Among many ET
products with global coverage and different level of uncertainties, WaPOR data sets are reported to have low biases and

good spatial variability across Africa (Blatchford et al., 2020; Weerasinghe et al., 2020).

In this study, we compare the model simulated ET, E, T and NPP to the corresponding level 1 (250 m) WaPOR data sets
(https://wapor.apps.fao.org/home/WAPOR_2/1) for the northern part of Morocco. The availability of E and T provides an
opportunity to explore the contribution of data assimilation on different components of ET, as the updated fields by SSM-
DA and LAI-DA may affect E and T in different ways. All the fields are spatially and temporally aggregated to 0.05° and a

monthly scale, respectively, for analysis between WaPOR data sets and the simulations.

### 2.5.2 FLUXCOM AND FLUXSAT GPP

The impact of LAI-DA on carbon fluxes is also evaluated by comparing the simulated GPP against the GPP product from the
FLUXCOM (Tramontana et al., 2016) and FLUXSAT (Joiner et al., 2018) projects. The FLUXCOM project uses machine
learning-based regression tools to upscale daily carbon flux estimates from flux tower sites into global gridded GPP

estimates covering the period of 2003-2015. The predictor variables required by the machine learning algorithms are based
exclusively on high resolution remote sensing data, including MODIS-based land cover and vegetation information and
ERA-Interim meteorological forcing variables. As the FLUXCOM products do not cover the SMAP period, we also utilize
the recently developed FLUXSAT GPP estimates, which are available from 2000-2020. Unlike many light-use efficiency
(LUE)-based models, the FLUXSAT GPP estimates do not use an explicit parameterization of LUE that reduces its value

from the potential maximum under limiting conditions such as temperature and water stress. Although the algorithm is
relatively simple, FLUXSAT took advantage of satellite-based SIF data to identify areas of high productivity and has been
shown to perform comparatively well compared to the FLUXCOM product. Similar to the comparison against WaPOR data,
we aggregate both products to monthly means at 0.05° spatial resolution for evaluation.


### 2.5.3 GOME MetOp-A SIF

Satellite-based sun-induced chlorophyll fluorescence (SIF) provides a new opportunity to monitor GPP for terrestrial ecosystems. As part of the vegetation photosynthesis process, the variation of SIF emitted by plants can be used to infer the actual functional state of the photosynthetic apparatus, since photosynthetic efficiency affects the efficiency of fluorescence emission (Rossini et al., 2015). In this study, the latest version (v28) of the monthly SIF product from the Global Ozone Monitoring Experiment-2 (GOME-2) aboard the MetOp-A satellite (Guanter et al., 2014; Joiner et al., 2013), from the period

of February 2007 to February 2019, is used to investigate its correlation with simulated GPP. Simulations are aggregated to 0.5° spatial resolution and averaged to monthly means for comparison.

### 2.5.4 Evaluation metrics

Statistical skill metrics include the Pearson's correlation coefficients for the (R) and anomaly (anomaly R) values based on monthly time series with 95% significance tested using Fisher's z transform test (Fisher, 1921), the root-mean-square

difference (RMSD) and bias (BIAS) with 95% significance tested by paired sample t test with temporal correlation accounted (Entekhabi et al., 2010). The anomaly R is calculated by removing the seasonal cycle from the time series, where the seasonal cycle is calculated as the multiyear average of each calendar month. R and anomaly R are used to examine the overall mismatch between the observations and simulations in terms of seasonality and interannual variability, respectively, while RMSD and BIAS provide information on how the simulations capture the magnitude of the fluxes. For the evaluations

conducted without SSM-DA involved, the longer time window (i.e., 2003-2019) allows us to further investigate how the simulations capture the interannual variability for each specific month. In these cases, R and RMSD are calculated for each month, separately, for a given time period depending on the observation-based dataset (e.g., 2009-2019 for WaPOR ET, E, T and NPP datasets). Correlation coefficients (R and anomaly R) are not additive measures and thus cannot be simply averaged, thus the median was computed as an evaluation score to represent the averaged performance for the full domain or

for the actively irrigated area. In addition, these analyses are stratified by major land cover types and different levels of irrigation intensity in order to quantify the impact of land cover types and irrigation intensity on the performance of data assimilation.

### 3. Results and Discussion

### 3.1 SMAP period evaluations

### 3.1.1 For the full domain

The four metrics (R, anomaly R, RMSD, and BIAS) were computed between the simulated monthly fluxes and the corresponding reference datasets for the period 2015-2019, as the SMAP soil moisture retrievals become available starting



from Mar 2015. Simulated E, T, ET, and NPP are compared against FAO WaPOR data sets while simulated GPP is compared against FLUXSAT GPP estimates.

Table 2 shows the overall performance for the OL, LAI-DA and SSM-DA simulations (no irrigation applied), and Figure 2 demonstrates the differences the DA simulations and OL masked using the applied significance test. For evaporation, assimilating soil moisture and LAI led to opposing impacts in terms of R. Slight degradations are found in LAI-DA (0.37) as compared to OL (0.38), while there is small improvement from SSM-DA (0.46). Nonetheless, the differences are only significant for a few grid cells in terms of both R and anomaly R, suggesting that both forms of data assimilation have

limited impacts on the temporal variability of evaporation (Figure 2 (a, f, k, p)). Over this region, all simulations produce much larger evaporation than the FAO WaPOR estimation, and the differences among the simulations are relatively small in terms of both RMSD and BIAS.

The simulated temporal variation for transpiration agrees better with the WaPOR estimates than that for evaporation, as the overall R and anomaly R for T are much higher than that for E in the OL simulation. For transpiration, assimilating LAI

greatly improved both R and anomaly R (Figure 2 (b, l)), and reduced RMSD with the percent change over 20% as compared to OL (Table 2). These positive impacts are mainly located in the vegetated north-western part of Morocco. However, SSM-DA failed to provide any skill in simulating transpiration as it led to degradation in terms of both R and anomaly R. LAI-DA tends to reduce the magnitude of transpiration, leading to larger negative BIAS, possibly contributed to by the smaller LAI magnitude in the observations as compared to that simulated by OL.

As LAI-DA and SSM-DA differ in altering the temporal variation as well as the magnitude of evaporation and transpiration, the impact on total ET is quite mixed. In general, for LAI-DA, improvements were found along the western coastal area, and degradations were found along the north-eastern coastal area in terms of R, while there is no significant impact in terms of anomaly R (Figure 2 (c, m)). For SSM-DA, the degradation in the seasonality of ET is limited to the north-western coastal area, while the degradation in interannual variability expands almost over half of the study domain (Figure 2 (h, r)), mainly

contributed by the degradation in transpiration.

The impact of LAI-DA and SSM-DA on NPP and GPP is similar to their respective impacts on transpiration. LAI-DA led to significant improvement on the two carbon fluxes in terms of R, anomaly R and RMSD while SSM-DA resulted in overall degradation. Consistent with its impact on T, LAI-DA tends to reduce the magnitude of both NPP and GPP as compared to OL, but the absolute BIAS is reduced for NPP as compared to FAO WaPOR while BIAS is increased for GPP as compared

to FLUXSAT. We note that both FAO WaPOR and FLUXSAT data sets are remote-sensing model data-driven products and are thus subject to uncertainties. However, the consistent results obtained by comparing the carbon fluxes against the two independent data sources highlight the benefit of assimilating LAI into the system.

### 3.1.2 Stratified by land cover types

For semi-arid and arid environments such as Morocco, land cover can be quite heterogeneous with interspersed agricultural

and natural vegetated areas. Therefore, land cover may play a large role on affecting the quality of the satellite derived soil


moisture and LAI estimates, thus affecting the data assimilation result. In this section, we analyze in depth the impact of land cover on the performance of data assimilation. Three major land cover types are selected for the analyses, which are "Open shrublands", "Croplands" and "Grasslands", covering almost all the vegetated areas of Morocco (Figure 1(a)).

As shown in Figure 3, the model demonstrates better skill in simulating evaporation for open shrublands and grasslands than
for croplands. However, the situation is the opposite when it comes to carbon fluxes. OL produces much higher correlation for both NPP and GPP for croplands. The relative performance for the OL, LAI-DA and SSM-DA simulations are similar across the three major land cover types. LAI-DA led to slight improvements in T and NPP and slight degradations in E whereas SSM-DA does the opposite. The overall correlation of ET is increased over croplands with LAI-DA, partly because the magnitude of T (17.8 mm/mo) is comparable to E (17.4 mm/mo) so that the degradation on evaporation has limited
impact on its skill in improving ET. Conversely, for open shrublands and grasslands, the benefits of LAI assimilation for transpiration are much weaker than that for croplands, as the magnitude of T/E ratio is lower than 30% for both land cover types. Therefore, LAI-DA does not provide comparable skill in improving ET.

The improvement in the simulation of transpiration and carbon fluxes by LAI-DA is largely due to the adjustment of the amplitude and phase of vegetation growth. Figure 4 shows the average monthly time series of LAI from the simulations,
along with the distribution of the months when LAI reaches the peak for all the grid cells per year stratified by land cover types. The time series indicates that OL significantly overestimates LAI, which is corrected by LAI-DA. In addition, LAI-DA leads to significantly different interannual variability. Comparatively, SSM-DA has limited impact on the evolution of LAI as compared to the OL run. The mismatch in terms of interannual variations is even more obvious under drought conditions. For instance, both OL and SSM-DA are not able to reflect the change of vegetation conditions in response to the
2015-16 drought event for croplands, while LAI-DA shows a clear reduction in the evolution of LAI during the 2015-16 growing season as compared to the adjacent years (Figure 4 (c)). Further, assimilating LAI also leads to changes to the phase of the LAI seasonality. In general, LAI-DA yields a peak in the LAI seasonality 1-2 months earlier than the OL and SSM-DA for all three land cover types (Figure 4 (b,d,f)).

### 3.2 Evaluating beyond the SMAP time period (2003-2019)

As the overall impact of LAI-DA is much greater than SSM-DA, especially for transpiration and carbon fluxes, we further extend the evaluation period beyond 2015-2019 for LAI-DA to investigate its impact on a long-term basis and to quantify its contribution across seasons. Interannual correlation and RMSD of evaporation and transpiration are calculated for each month, and Figure 5 shows the median of each metric stratified per land use type.

The model in general provides much higher correlation for transpiration than for evaporation in winter and spring,
overlapping with the growing seasons, regardless of data assimilation and land use types. However, it generally overestimates transpiration during summertime, especially for croplands, likely due to the misrepresentation of the vegetation seasonality. Assimilating LAI into the model improves the interannual correlations for transpiration across all land use types and all seasons and generally reduces RMSD for croplands. In the case of evaporation, LAI-DA has marginal





impact on correlation but leads to larger RMSD, indicating less agreement in representing the magnitude of E, which may
stem from differences in the ET algorithm and the associated uncertainties from both Noah-MP and WaPOR.

Besides the ET components, we evaluate the impact of LAI assimilation on carbon fluxes by comparing the NPP and GPP
estimates against reference data sets. Similar to Figure 5, Figure 6 shows the interannual correlation and RMSD for NPP
evaluating against the WaPOR dataset (Figure 6 (a,e)) and for GPP evaluating against the FLUXSAT (Figure 6 (b,f)) and the
FLUXCOM (Figure 6 (c,g)) GPP estimates and the GOME-2 SIF estimates (Figure 6 (d)). As the impact of LAI-DA on the
carbon fluxes are similar among the three land use types (not shown), medians of the metrics for the sum of these major land
use types are shown to represent the overall performance. In general, the highest correlation for NPP is found in summer and
winter time while the highest correlation for GPP aligns with the peak growing season of Dec-Apr. However, the largest
RMSD values for both NPP and GPP also occur within the growing season (Feb-Apr). The results suggest that assimilating
LAI consistently improves the interannual variability of both NPP and GPP for all months and the greatest improvements are
found within the growing seasons. Moreover, LAI-DA is able to reduce approximately half of the RMSD for both carbon
fluxes.

Evaluating against multiple independent datasets provides different insights on the impact of data assimilation on the
performance of both energy and carbon fluxes. The overall accuracy of estimated ET components, NPP and GPP, both
within and beyond the SMAP time period, suggest that LAI-DA has a greater beneficial impact than SSM-DA. This is
consistent with the (Kumar et al., 2020) study, which demonstrated that updating vegetation phenology is more effective for
generating improvements in evaporative fluxes.

### 3.3 The impact of irrigation

Although precipitation serves as a primary source for plant transpiration and soil evaporation, irrigation can also play a role
in supplying water for agricultural productivity and has a significant contribution to enhance the ET and carbon fluxes. To
quantify the impact of irrigation on fluxes, we investigate performance of the second set of experiments (i.e., $OL_{irr}$, LAI-
$DA_{irr}$ and SSM-$DA_{irr}$) with a special focus on the grid cells that are actively irrigated. Comparing $OL_{irr}$, LAI-$DA_{irr}$, and SSM-
$DA_{irr}$ with the OL simulation enables us to examine how irrigation affects the simulated fluxes under the original, the LAI or
soil moisture assimilation configuration.  Note that according to the irrigation rules applied in the model, the growing season
time window defined by GVF threshold and the root zone soil moisture can both alter the irrigation timing and amount, thus
affecting the associated fluxes. Similar to the analyses in previous sections, we first evaluate the irrigation impact within the
SMAP time period (Table 3) and then we exclude the SSM-DA simulation and evaluate the impact beyond the SMAP time
period (Figure 7).

Table 3 shows the overall performance for the OL, $OL_{irr}$, LAI-$DA_{irr}$ and SSM-$DA_{irr}$ simulations for actively irrigated areas.
Interestingly, $OL_{irr}$ dramatically improves the correlation for evaporation, bringing the median correlation from -0.01 in OL
to 0.68 and SSM-$DA_{irr}$ further improves the correlation to 0.71. They together indicate that the inclusion of irrigation is the
major factor contributing to the improvement of the seasonal variation of evaporation and assimilating surface soil moisture





leads to further improvements. The two simulations slightly degrade anomaly R while LAI-DA$_{irr}$ has almost no impact on either. Compared to OL, including irrigation with or without data assimilation increases the positive BIAS for evaporation. Conversely, OL$_{irr}$ and SSM-DA$_{irr}$ degrades R and anomaly R for transpiration, NPP, and GPP while LAI-DA$_{irr}$ leads to

significant improvements. Note that the improvement in T, NPP, and GPP is mainly attributed to assimilating LAI. The inclusion of irrigation provides further improvements but is relatively small when comparing LAI-DA with LAI-DA$_{irr}$ for actively irrigated area (not shown). The overall BIAS of total ET and NPP is the smallest in LAI-DAirr while both OL$_{irr}$ and SSM-DA$_{irr}$ largely increases the BIAS for these two terms, as well as that for GPP.  It is also interesting to note that the sign of BIAS for transpiration is different among the simulations. OL underestimates transpiration and LAI-DA$_{irr}$ further

increases this dry BIAS as compared to WaPOR, whereas both OL$_{irr}$ and SSM-DA$_{irr}$ lead to increased transpiration, resulting in positive transpiration BIAS.

The reason that both OL$_{irr}$ and SSM-DA$_{irr}$ lead to improved seasonal evolution of evaporation compared to OL is that irrigation is erroneously triggered during summertime as the period is identified as within the growing season according to the dynamic phenology module. To bring the dry root zone soil moisture to field capacity, a large amount of water is

therefore applied into the effective root zone soil layers, producing a large peak of evaporation, which is also represented in the WaPOR datasets. OL failed to simulate the summer peak of evaporation likely due to the limited water availability for deep soil evaporation, as the soil thickness for Noah-MP is only 2 m, and/or the underestimation of the root water uptake and groundwater capillary rise. LAI-DA$_{irr}$ does not provide improvements because irrigation is not triggered during summertime, which falls outside of the growing season indicated by the assimilated LAI observations with an early peak LAI seasonality.

In cases like this, the improvement in evaporation in OL$_{irr}$ and SSM-DA$_{irr}$ is related to the fact that the erroneously applied irrigation compensates for the model structural error. It does improve the correlation for E, but it also leads to degradation in BIAS for ET and carbon fluxes.

For the evaluation beyond the SMAP time period (spanning portions of 2003-2019), we further stratify the actively irrigated areas into three classes based on the irrigation fraction intensity (IRfrac): (i) the lightly irrigated area, where the irrigation

fraction is lower than 25%; (ii) the moderately irrigated area, where the irrigation fraction is between 25% and 50%; and (iii) the heavily irrigated area, where the irrigation fraction is higher than 50% (Figure 2 (b)).

Medians of the interannual correlations for E, T, NPP and GPP for the three classes are shown in Figure 7. It is interesting to note that when compared with OL, OL$_{irr}$ degrades the correlation for E, T and GPP for almost all months and the degradation becomes larger as irrigation fraction intensity increases. Interacting with the original dynamic phenology module, applying

irrigation does not provide any skill in improving the interannual variability for each specific month, but rather makes it worse. Previous studies (Liu et al., 2016; Niu et al., 2020) have demonstrated that the dynamic phenology module in Noah-MP produces large errors in seasonal evolution of vegetation phenology, possibly due to the overly simplified parameterization of growth characteristics and the stomatal response to stresses. In addition, many of the parameters and scaling factors determining water availability are derived and calibrated within CONUS and may not be optimal for

application to the MENA region. The misrepresentation of vegetation condition may introduce erroneous information to





trigger irrigation, and irrigation further affects the vegetation growth and the associated carbon fluxes by altering the soil moisture condition and the associated water stress controlling factor. Therefore, errors accumulate and drive the simulated fluxes further away from the reference datasets.

In this context, assimilating LAI can correct for the seasonal evolution of vegetation conditions, thus constraining the time
window to trigger irrigation, leading to improved correlation for transpiration, NPP and GPP. However, its impact on evaporation is very limited. Temporally, its contribution would be centered within the growing season so that it is not likely to compensate for model weakness in simulating the summertime evaporation; spatially, significant differences occur mostly in intensely irrigated areas, where transpiration is the dominant component in ET.

## 3.4 Implications for drought categorization

### 3.4.1 Applications for root zone soil moisture – based drought

Limited by the availability of in situ soil moisture observations, it is challenging to directly evaluate the data assimilation impact on soil moisture. Nevertheless, we can quantify how data assimilation differentiates the categorization of drought and reproduces the evolution, duration, and intensity of past drought events as an indirect way to evaluate its impact on root zone soil moisture.

Estimates of droughts are generated through percentile-based indices using root zone soil moisture outputs (Top 1 m depth) from OL, SSM-DA and LAI-DA. The percentile-based root zone soil moisture indicator is computed in a manner similar to that used in the NLDAS drought monitoring system (Kumar et al., 2016; Sheffield et al., 2012). Daily outputs from OL and LAI-DA for the period of 2002-2019 are used to generate the climatology and the daily percentile values are computed by ranking each day's estimates against the climatology for OL and LAI-DA respectively. Since SSM-DA involves a shorter
time period (2015-2019), we use the climatology generated by OL to rank the estimates from SSM-DA without further scaling, as the SMAP observations are already scaled to OL before the assimilation is performed. The drought percentage area values are then produced per land cover type and are categorized into five drought levels: D0 (abnormally dry, percentile $\leq$ 30%), D1 (moderate drought, percentile $\leq$ 20%), D2 (severe drought, percentile $\leq$ 10%), D3 (extreme drought, percentile $\leq$ 5%), and D4 (exceptional drought, percentile $\leq$ 2%).

Figure 8 presents the time series of monthly drought percentage area derived from OL, LAI-DA and SSM-DA for five drought percentile categories, stratified by the three major land cover types over the SMAP period (2015-2019). Overall, all simulations are able to capture major drought periods for Morocco, such as the 2015-16 (Bijaber et al., 2018) and 2018-19 (Bhaga et al., 2020) events. However, the categorized drought intensity and corresponding duration vary. For example, for the Dec 2015 – Feb 2016 drought event over open shrublands, LAI-DA estimates 28% less area experiencing extreme
drought (D3) as compared to the OL run. Moreover, LAI-DA also tends to estimate a weaker drought evolution and faster recovery during the post-drought period, while SSM-DA differs from OL and LAI-DA more in terms of the drought intensity categorization, as 14% of area is detected as the exceptional drought (D4) compared to zero in both OL and LAI-





DA. For the 2018-19 event, under which cereal production is reported to have decreased 49% compared to the 2017-18 season, LAI-DA tends to estimate more severe drought intensity and longer duration for croplands but relatively weaker and

smaller expansion for open shrublands as compared to both OL and SSM-DA. This may imply that the incorporation of the LAI observations helps to reflect different drought representation across land cover types, as agricultural lands seem to be more vulnerable facing a severe drought while natural vegetation types are less affected. However, without LAI-DA, the model is not able to distinguish these drought sensitivities. Moreover, LAI estimated in OL is much higher than that simulated in LAI-DA (Figure 4) especially for open shrublands and grasslands, leading to reduced transpiration through

vegetation and drier root zone soil moisture. Assimilating LAI helps to correct this overestimation of drought by improving the magnitude of LAI simulation. This is also consistent with the findings in Mocko et al. (2021) for the CONUS region.

To further assess the impact of data assimilation on differentiating the categorization of drought as compared to OL, Figure 9 shows the percentage of area under each drought level for LAI-DA and SSM-DA against OL stratified by the three major land cover types. All three simulations are similar in diagnosing the extent of the mild drought (D0), especially for open

shrubland and croplands. More differences are seen in categorizing the moderate to severe drought events (D1-D2) and there is no clear pattern associating with the differences. When it comes to the extreme and exceptional drought (D3-D4), LAI-DA and SSM-DA show the opposite tendency as compared to OL for open shrublands and grasslands, in that LAI-DA tends to limit the spatial extent of the extreme and exceptional drought while SSM-DA is more likely to expand the impact of higher level of drought extremes. However, their tendency is similar over croplands. It should be noted that uncertainties exist in

assigning the percentile for SSM-DA based on the climatology of OL. Nevertheless, the result that LAI-DA tends to limit estimates of the most severe drought categories implies that assimilating vegetation states may have a stronger impact on simulation of extreme moisture anomalies, information on which might not be carried by the soil moisture observations or represented by the model.

### 3.4.2 Data assimilation impact on vegetation response to drought

As demonstrated in the above section, vegetation may contain critical information in altering the root zone soil moisture-based drought classifications, thus we further investigate how the model simulates vegetation in response to drought. Representing vegetation response to drought is often a challenge in land surface modeling as most of the models use oversimplified parameterizations to downregulate stomatal conductance and photosynthesis under drought stress (Eller et al., 2020; Liu et al., 2020; Niu et al., 2020). In Figure 10 we examine the spatial distribution of LAI anomaly under the 2015-16

drought by comparing the difference between LAI in Feb 2016 and its climatology derived within the SMAP period (2015-2019). The vegetation response to drought in the OL estimates significantly differs from the observation, as the OL underestimates the spatial variability of the response, the pattern of which may relate to land cover types. More specifically, OL tends to underestimate the LAI anomaly over the northwestern region including most part of the croplands while overestimating the drought effect for the northeastern and along the southern edge of the open shrublands. The spatial pattern

of the LAI anomaly in the SSM-DA is similar to that of the OL, except for the southeastern area, where SSM-DA brings the





positive LAI anomaly into closer agreement with the observation. However, the increased small-scale variability of the LAI anomaly in the SSM-DA likely reflects the fact that the assimilated surface soil moisture has degraded the model's ability to simulate vegetation conditions in terms of spatial consistency. Although assimilating soil moisture inherits uncertainties due to shallow vertical penetration depth and scale mismatch, this does not necessarily mean that the observed soil moisture
condition provides limited or erroneous stress information to the model. The fact that SMAP observation is scaled to model via CDF-matching also means that observational information is lost so that we are only incorporating anomalies outside of what is captured by precipitation (Nearing et al., 2018), limiting the possible added value via SSM-DA. Besides, the simplified concepts and parameterization of stomatal conductance and dynamic phenology schemes may also fail to reasonably digest and properly apply the stress information to represent vegetation response to drought. In this case, LAI-
DA, as expected, can reasonably replicate the spatial distribution of vegetation in response to drought by constraining the vegetation cycle, thus leading to more accurate simulation of transpiration and associated carbon fluxes.

## 4. Conclusions

Morocco is known to have experienced intensified drought events during recent decades and this increasing trend, associated with global climate change, will likely be more evident in the future (Verner et al., 2018). In fact, many countries in the
MENA region are vulnerable to drought due to underlying aridity and limits to current water and agricultural management practices, as well as the limited information available to aid decision making for drought preparedness. A robust drought monitoring and early warning system would be beneficial to mitigate drought effects and facilitate timely and effective responses from government and private sector stakeholders (Fragaszy et al., 2020). However, challenges exist as modeling efforts in the MENA region are severely limited by the lack of in situ observations to support an optimal set of
parameterizations. Remotely sensed observations of soil moisture and vegetation conditions contain information from both anthropogenic and natural changes in response to climate variability and extremes, and data assimilation provides a way to incorporate such information into land surface modeling, extending the potential of benefiting drought monitoring and forecasting efforts.

In this study, we look at Morocco as an example to demonstrate the capabilities of remotely sensed soil moisture and leaf
area index (LAI) to improve the simulation of water-energy-carbon fluxes and the representation of drought conditions over the MENA region. The combination of GDAS and IMERG meteorological forcing fields is selected to drive the modeling system and the EnKF algorithm is used to separately assimilate the SMAP-based surface soil moisture retrievals and MODIS-based LAI retrievals into Noah-MP during the 2015-2019 and 2002-2019 time periods, respectively. We conducted two sets of simulations (with or without irrigation) to investigate the influence of data assimilation and its interaction with
irrigation via informing the dynamic phenology on the estimation of ET components and carbon fluxes by comparing against multi-source satellite observations.



Results show that assimilating soil moisture does not meaningfully improve model representation of hydrological processes for the study region but rather leads to degradation in the simulation of interannual variations of T, NPP, and GPP. In contrast, assimilating LAI leads to substantial improvements to simulation of these fluxes in terms of both temporal variation

and RMSE, which are primarily due to the correction of the phase and magnitude of LAI. Both SSM-DA and LAI-DA have limited impact on evaporation in terms of the seasonal and interannual variability, but they differ from OL in the partitioning of E and T. SSM-DA tends to increase both E and T while LAI-DA leads to increase E but decrease T, likely due to the reduced magnitude of LAI. The fact that the two data assimilation experiments affect E and T differently results in mixed skill on the estimation of ET. For open shrubland and grassland where the ratio of E/T is high, both SSM-DA and LAI-DA

have limited impact on ET while ET is improved for croplands by LAI-DA as the magnitude of T is comparable to E.

In the presence of irrigation, significant improvements in the seasonal cycle of evaporation are observed in both OL$_{irr}$ and SSM-DA$_{irr}$. This is, however, a result of irrigation water supply erroneously capturing the peak during the summertime. The failure to capture the summertime E peak in OL is attributed to model limitations in describing the water availability and underestimation of groundwater capillary rise. Therefore, the erroneously applied irrigation compensates for the model

structural error. Though it improves the correlation for E, the magnitude of E is greatly overestimated. In addition, the irrigation simulation is also found to degrade the temporal variability of carbon fluxes. In this case, LAI-DA$_{irr}$ provides comparable skill to LAI-DA by constraining the timing of irrigation within the MODIS LAI observation informed growing season. The inclusion of irrigation provides marginal improvements in LAI-DA$_{irr}$ compared to LAI-DA specifically in the correlation for T, NPP and GPP compared to LAI-DA (not shown).

LAI-DA$_{irr}$ outperforms other simulations by correcting vegetation phenology processes, whereas SSM-DA$_{irr}$ provides no measurable skill in avoiding the erroneous triggering of summertime irrigation, introducing greater bias to other variables. This result underscores the fact that data assimilation helps to diagnose model weakness, and that it may amplify model errors by translating the change of one variable into changes of other variables without proper support from model physics, similar to the findings of other data assimilation applications (Girotto et al., 2017; Kolassa et al., 2017b).

The influence of SSM-DA and LAI-DA on categorizing drought is examined by generating percentile-based root zone soil moisture drought indicators. The percentage of area under drought for five drought severity categories is quantified for three major land cover types. Results suggest that both SSM-DA and LAI-DA do not differ much from OL for mild drought events, but the differences become greater as the drought severity category increases. For example, assimilating LAI tends to reduce the estimated area under D4 category for open shrubland and grassland, implying alleviated moisture anomalies

under extreme conditions in response to vegetation states which might not be captured by either OL or SSM-DA. This study has focused on differences between simulations with respect to drought indicators. Further assessment against independent data is needed to assess whether the differences introduced by data assimilation improve the accuracy of drought categorizations, or whether these changes offer a benefit for drought monitoring and management in MENA countries.

**Data availability**

The model output relevant to this work will be made available through the Johns Hopkins University Data Archive (https://archive.data.jhu.edu/) once the final version of this paper is published.

**Author contributions**

WN, CDP, SVK, KRA conceived the study. WN, KRA, and IEM set up the model. SVK, DMM, and MN provided support on modelling development. SPM provided support on input datasets. WN designed the workflow and conducted the

analyses, with all co-authors providing input. WN led the manuscript writing, with contributions from all co-authors.

**Competing interest**

The authors declare that they have no conflict of interest.

**Disclaimer**

This publication was made possible through the support of the Office of Technical Support, Bureau for the Middle East, U.S.

Agency for International Development, under the terms of Award No. 7200-ME-18-IO-00001. The opinions expressed in this publication are those of the authors and do not necessarily reflect the views of the U.S. Agency for International Development or the United States government.

**Acknowledgements**

Computational resources were provided by the NASA's Center for Climate Simulation (NCCS). Different data sets used for

model evaluation were obtained from various sources described in section 2. We would like to acknowledge the NASA LIS team for their help on model development and support of the GDAS and IMERG datasets. We also thank Dr. Timothy Lahmers and Dr. Kimberly Slinski for providing valuable feedback to this study.

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

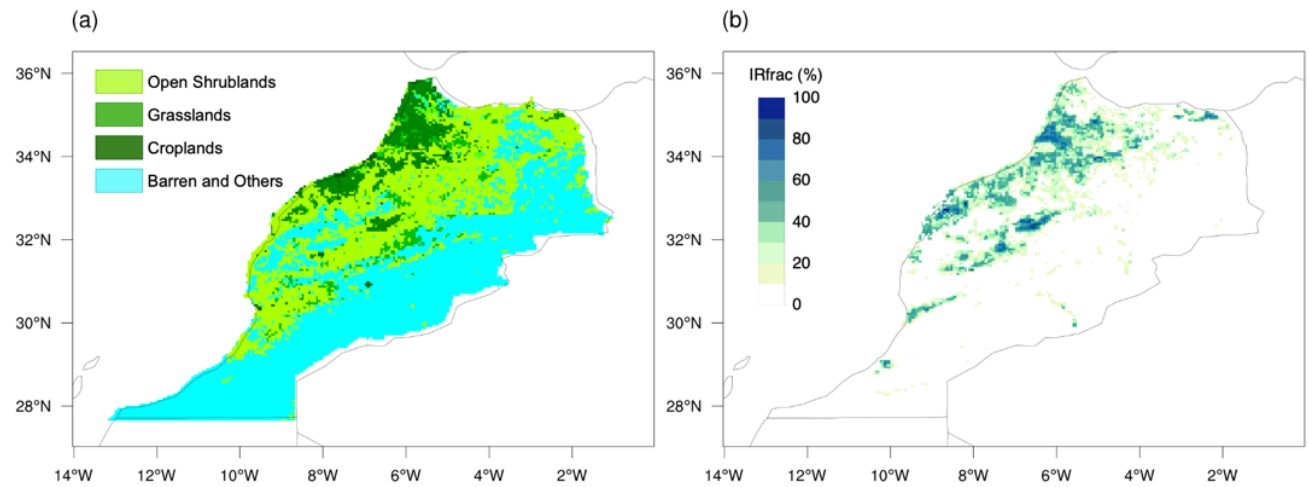

**Figure 1: Map of (a) the major land cover types based on MODIS-IGBP data set and (b) the composite irrigation fraction intensity for Morocco.**



Figure 2: The difference of correlation (*R*) – computed as DA minus OL for the period of 2015-2019 for (a-e) LAI-DA and (f-j) SSM-DA simulations in terms of evaporation (E), transpiration (T), evapotranspiration (ET), net primary production (NPP) and gross primary production (GPP). Panels (k-t) are the same, but for the difference of anomaly correlation (*anomaly R*). Red colors indicate that the assimilation improves *R* or *anomaly R* with respect to the OL and blue colors indicate a degradation at 95% significance level using Fisher's z transform test.




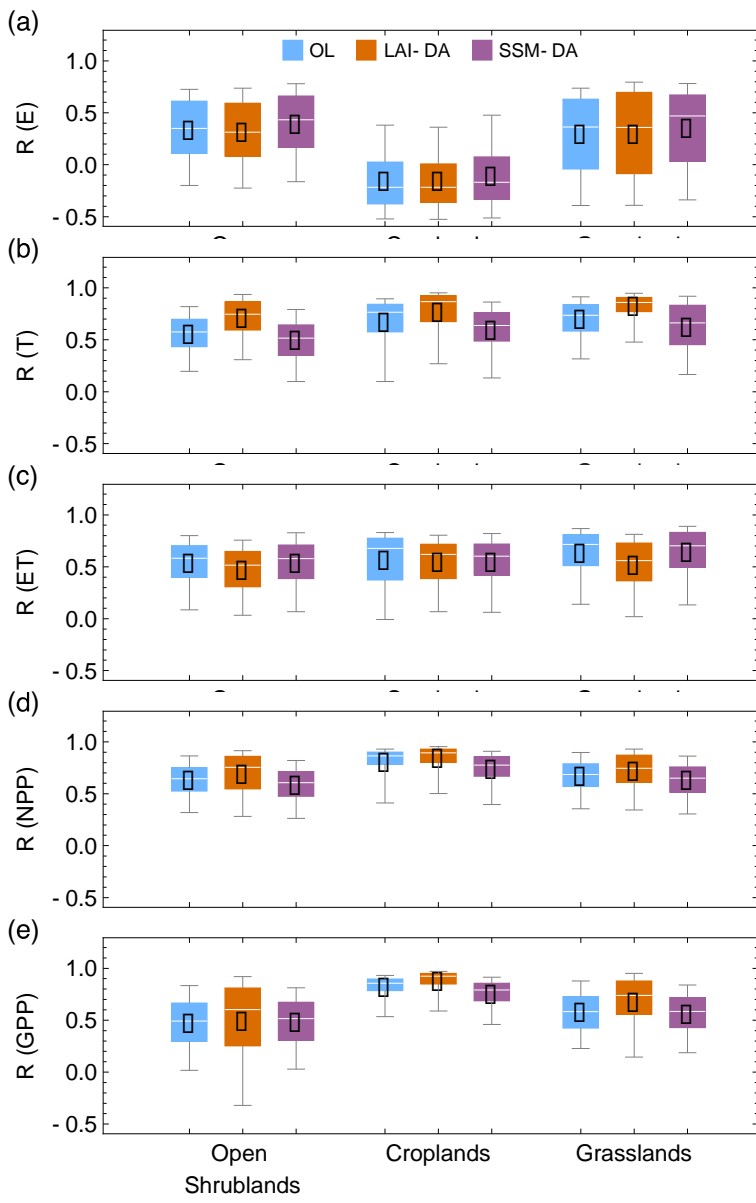

**Figure 3: Boxplots of the correlation for (a) evaporation (E), (b) transpiration (T), (c) evapotranspiration (ET), (d) net primary production (NPP), and (e) gross primary production (GPP) stratified by the major landcover classifications for the OL, LAI-DA, and SSM-DA simulations for the period of 2015-2019. White lines represent the median, boxes represent the upper- and lower-quantiles, black whiskers represent the 5- and 95-th percentiles, and black dots represent the mean values.**





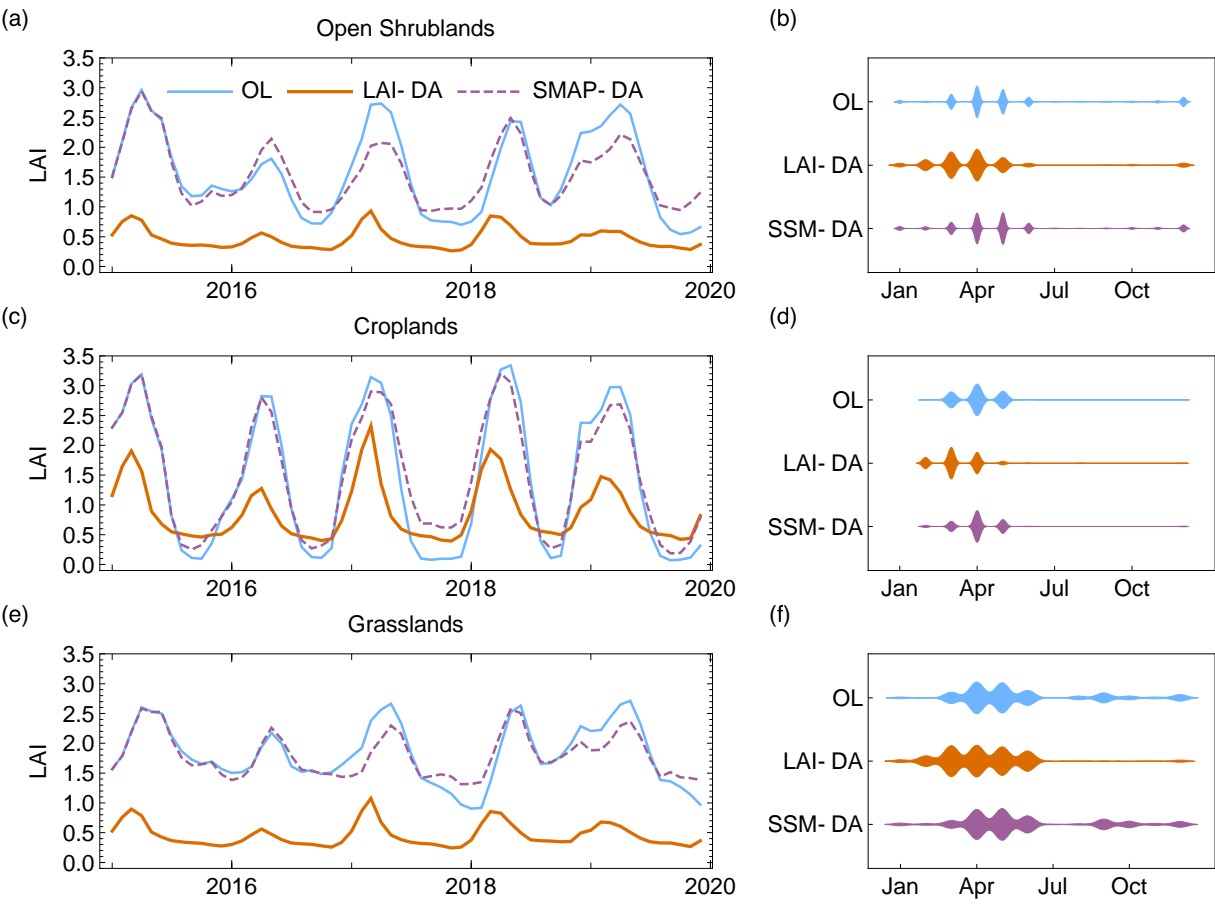

**Figure 4: Monthly time series of LAI estimated from OL, LAI-DA, SSM-DA and the corresponding distribution of peak LAI month for (a,b) Open shrublands, (c,d) Croplands, and (e,f) Grasslands.**





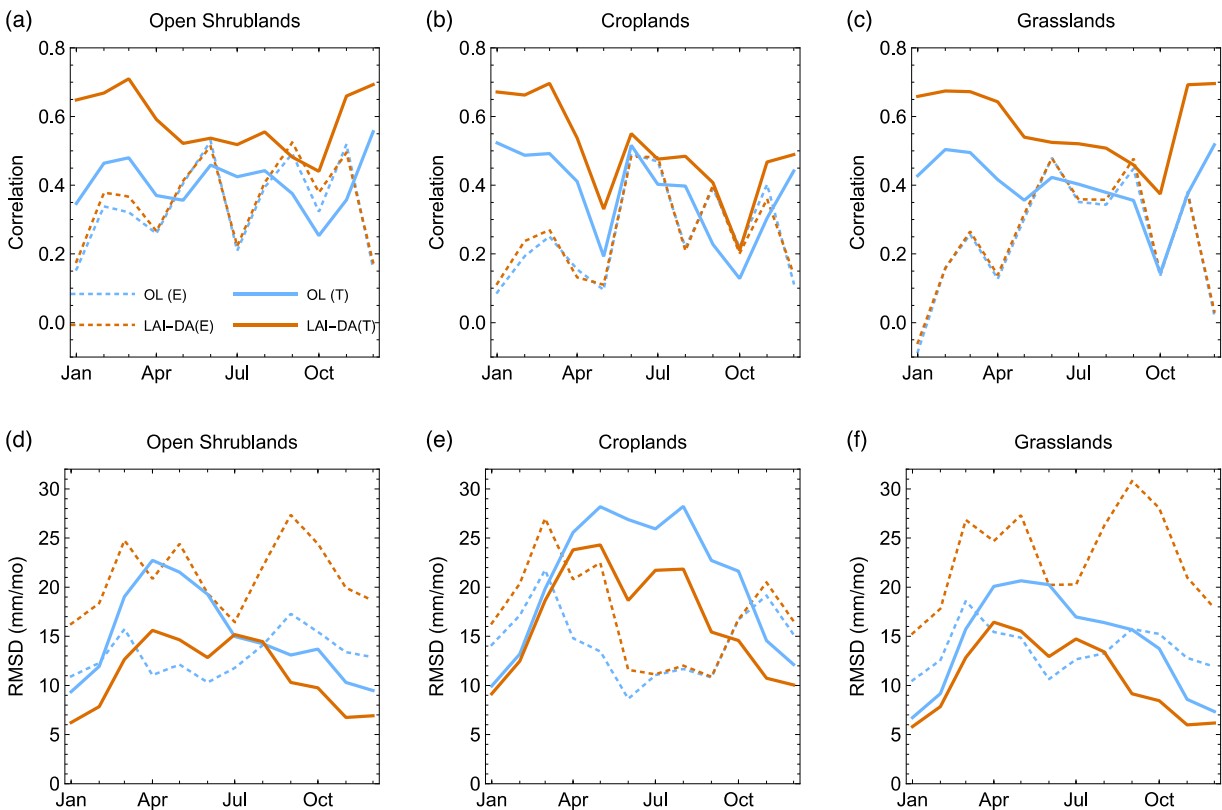

**Figure 5: Interannual (a-c) correlation and (d-f) RMSD of evaporation (E) and transpiration (T) averaged over the major land cover classifications for the OL and LAI-DA simulations for the period of 2009-2019.**





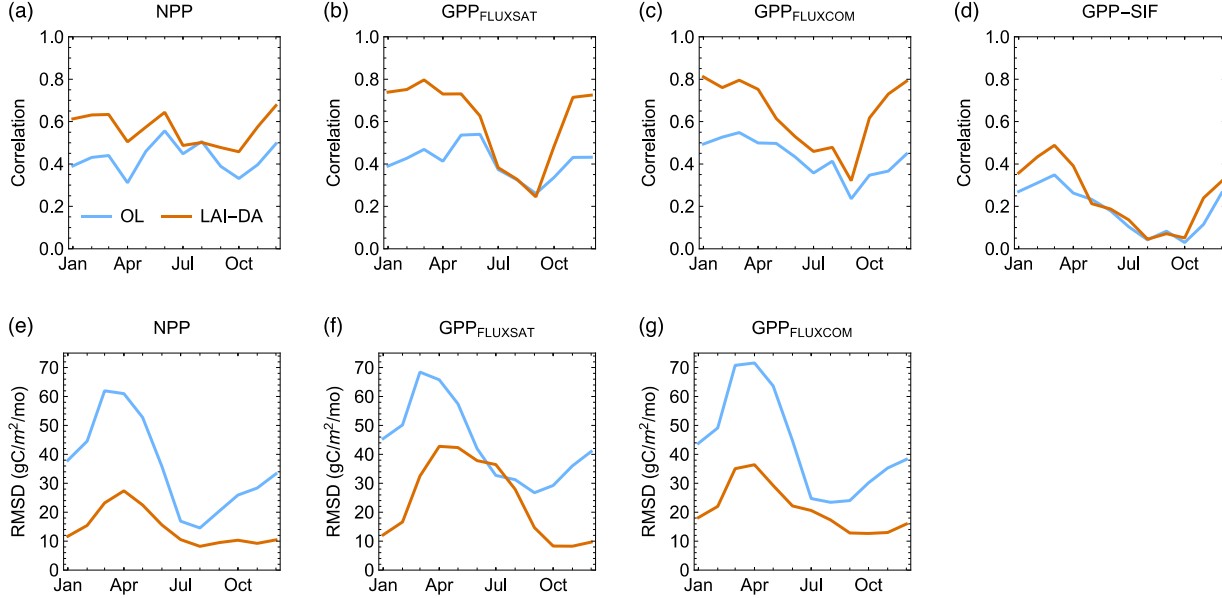

**Figure 6: Metrics for the OL and LAI-DA simulated net primary production (NPP) and gross primary production (GPP) against reference datasets averaged over the sum of open shrublands, croplands, and grasslands. Shown are (a) correlation and (e) RMSD for NPP evaluated against FAO WaPOR NPP data set for the period of 2009-2019; correlation (b, c) and (f, g) RMSD for GPP evaluating against FLUXSAT GPP product for the period of 2003-2019 and FLUXCOM GPP product for the period of 2003-2015; and (d) correlation between the simulated GPP and GOME-2 sun-induced chlorophyll fluorescence (SIF) for the period February 2007 to February 2019.**

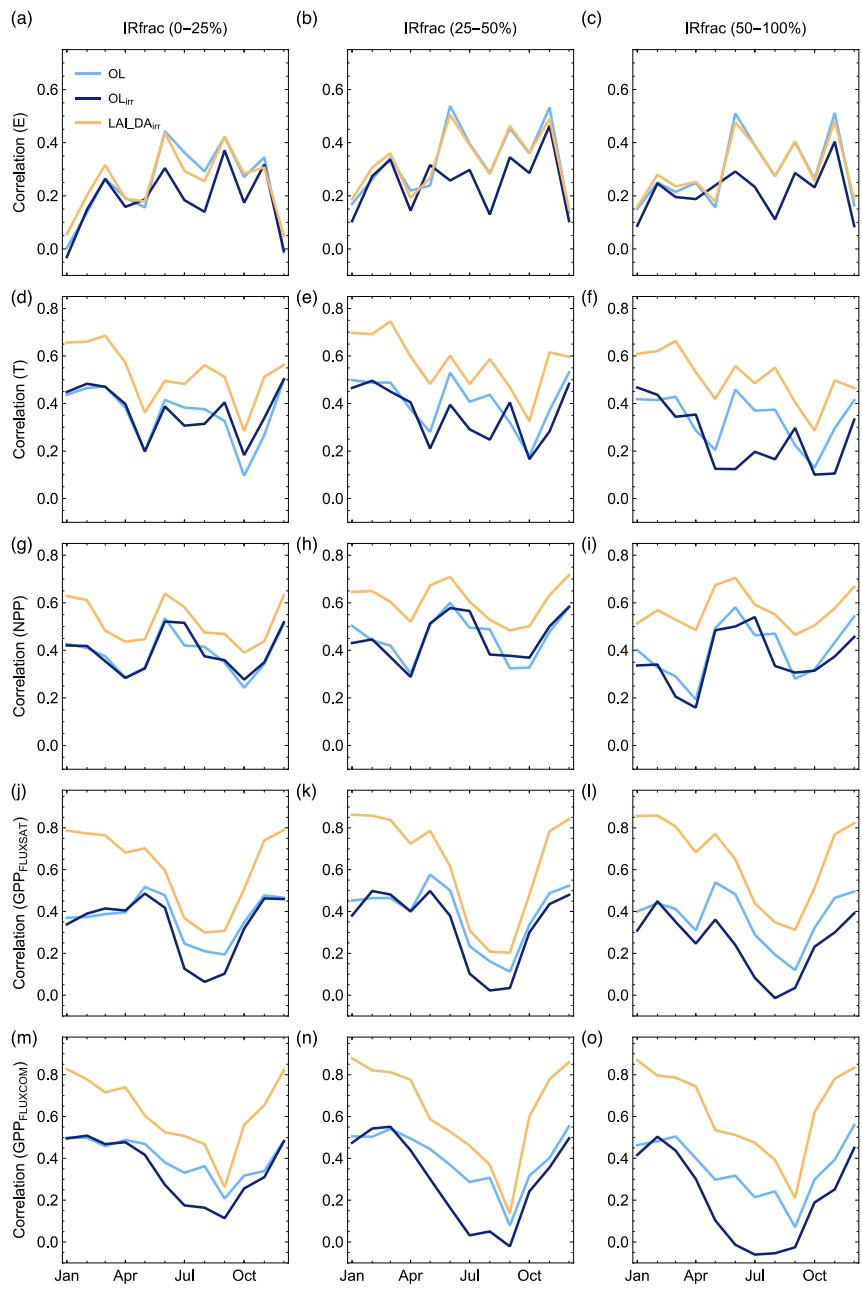

**Figure 7: Interannual correlation of evaporation (E) and transpiration (T), net primary production (NPP) and gross primary production (GPP) averaged over actively irrigated areas with low (left column), moderate (middle column) and high (right column) irrigation fraction intensities for the OL, OL$_{irr}$, and LAI-DA$_{irr}$ simulations.**


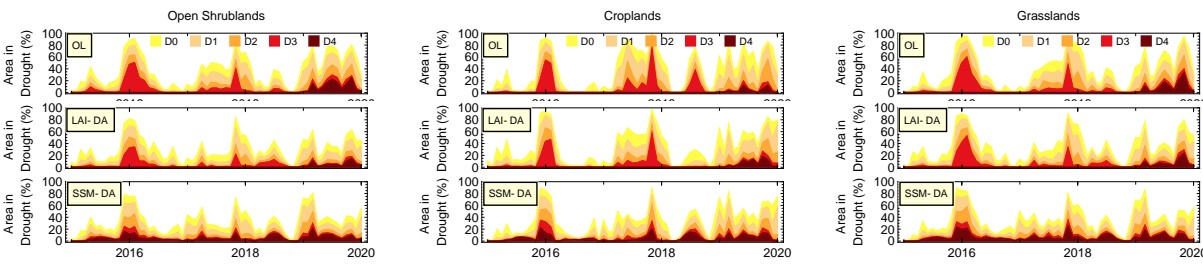

**Figure 8: Time series of percentages of area under drought in OL, LAI-DA, and SSM-DA for each root zone soil moisture drought percentile category and for each land use type.**

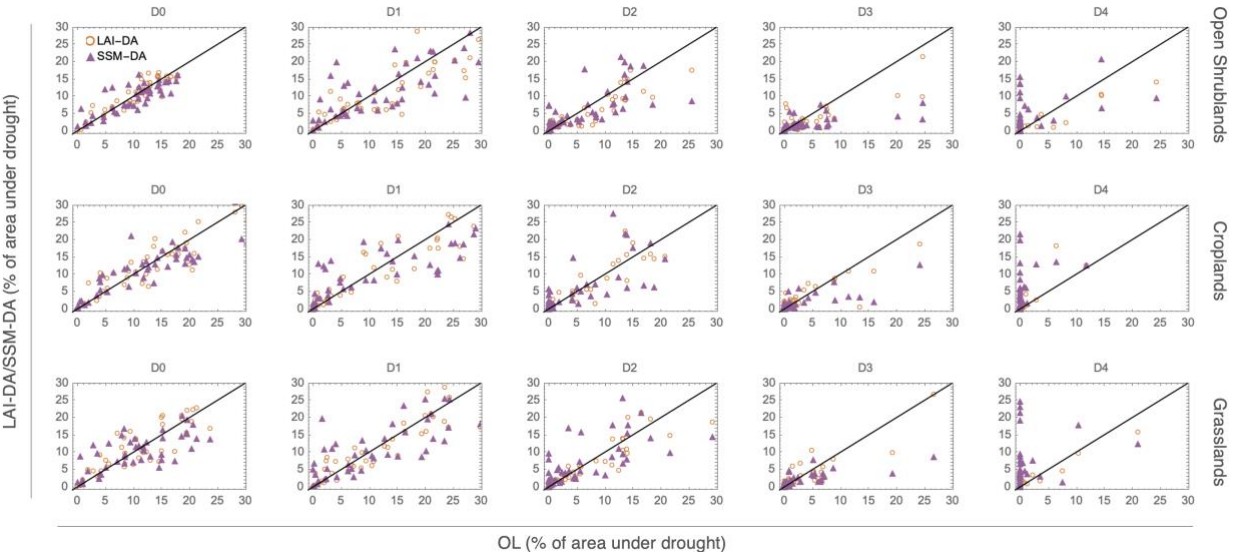

**Figure 9: The scatter plot for the percentage of area under drought between OL and LAI-DA/SSM-DA for the five drought categories stratified by the three major land use types. Data of summertime (Jun-Aug) are excluded for this analysis.**

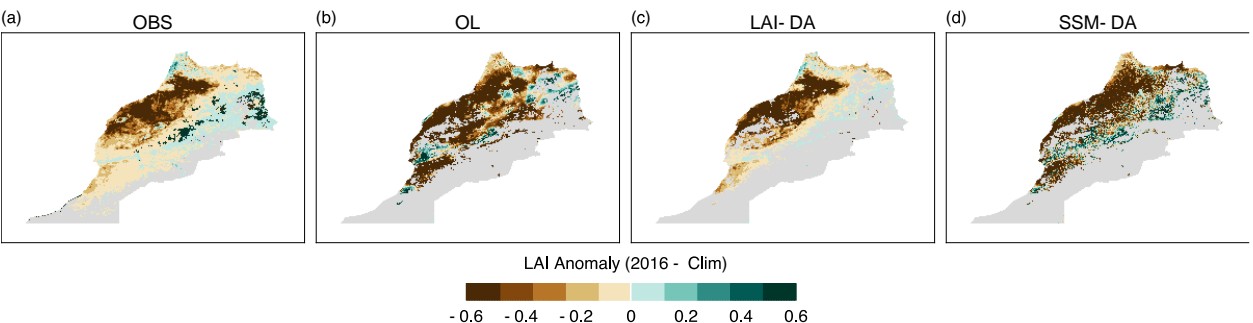

**Figure 10: LAI anomaly under a drought event (2016 Feb) with respect to climatology of Feb within SMAP period (2015-2019) for (a) MODIS observation, (b) OL, (c) LAI-DA, and (d) SSM-DA.**





825 **Table 1: Ensemble perturbation parameters in the assimilation simulations.**

| Variable | Type | Std dev | Temporal correlation | Perturbation cross-correlations | | |
|---|---|---|---|---|---|---|
| | | | | SW | LW | Precip |
| **Met- forcings** | | | | | | |
| SW | M | 0.2 (dimensionless) | 24 h | 1 | -0.5 | -0.8 |
| LW | A | 30 W m$^{-2}$ | 24 h | -0.5 | 1 | 0.5 |
| Precip | M | 0.5 (dimensionless) | 24 h | -0.8 | 0.5 | 1 |
| **LSM States for SSM-DA** | | | | | | |
| Surface SMC | A | 0.02 m$^3$ m$^{-3}$ | 1 h | | | |
| **LSM States for LAI-DA** | | | | | | |
| LAI | A | 0.01 m$^2$ m$^{-2}$ | 1 h | | | |

*Note*. Multiplicative (M) or additive (A) perturbations are applied to the meteorological ("Met") fields: incident shortwave radiation (SW), incident longwave radiation (LW), precipitation (Precip), and the model states: surface soil moisture content (SMC) and LAI.

**Table 2: Median evaluation metrics of monthly correlation (*R*), anomaly correlation (*anomaly R*), RMSD and BIAS for the OL, LAI-DA and SSM-DA simulations against FAO WaPOR datasets for evaporation (E), transpiration (T), evapotranspiration (ET), net primary production (NPP) and against the FLUXSAT data set for gross primary production (GPP) for the period of 2015-2019**
830 **over the Morocco domain.**

| | | Morocco Domain | | |
|---|---|---|---|---|
| | | OL | LAI-DA | SSM-DA |
| E (mm/mo) | *R* | 0.38 | 0.37 | **0.46** |
| | *anomaly R* | 0.34 | 0.35 | 0.36 |
| | RMSD | 13.22 | 14.95 | 12.15 |
| | BIAS | 10.31 | 13.87 | 11.9 |
| T (mm/mo) | *R* | 0.61 | **0.77** | 0.54 |
| | *anomaly R* | 0.44 | **0.66** | 0.28 |
| | RMSD | 12.32 | **7.16** | 12.74 |
| | BIAS | -0.01 | -8.41 | 0.71 |
| ET (mm/mo) | *R* | 0.51 | 0.49 | 0.53 |
| | *anomaly R* | 0.39 | 0.37 | 0.34 |
| | RMSD | 16.73 | 16.58 | 15.93 |
| | BIAS | 11.07 | 10.15 | 12.88 |
| NPP (g/m$^2$/mo) | *R* | 0.68 | 0.78 | 0.63 |
| | *anomaly R* | 0.48 | **0.64** | 0.31 |
| | RMSD | 27.64 | **11.58** | 26.6 |
| | BIAS | 17.01 | **-3.73** | 18.46 |
| GPP (g/m$^2$/mo) | *R* | 0.57 | **0.69** | 0.58 |
| | *anomaly R* | 0.51 | **0.8** | 0.37 |
| | RMSD | 38.33 | **19.91** | 36.19 |
| | BIAS | 12.26 | -16.02 | 14.94 |

Bold font indicates median improvement over 20%.





**Table 3: Median evaluation metrics of monthly correlation (*R*), anomaly correlation (*anomaly R*), RMSD, and BIAS for the OL, OL$_{irr}$, LAI-DA$_{irr}$ and SSM-DA$_{irr}$ simulations against FAO WaPOR datasets for evaporation (E), transpiration (T), evapotranspiration (ET), net primary production (NPP), and against the FLUXSAT data set for gross primary production (GPP) for the period of 2015-2019 over the actively irrigated area.**

|  |  | Actively Irrigated Area | | | |
|---|---|---|---|---|---|
|  |  | OL | OL$_{irr}$ | LAI-DA$_{irr}$ | SSM-DA$_{irr}$ |
| E (mm/mo) | *R* | -0.01 | 0.68 | 0.07 | 0.71 |
|  | *anomaly R* | 0.17 | 0.13 | 0.17 | 0.12 |
|  | RMSD | 11.14 | 8.05 | 12.47 | 8.76 |
|  | BIAS | 7.18 | 14.52 | 11.49 | 17.27 |
| T (mm/mo) | *R* | 0.62 | 0.5 | 0.78 | 0.49 |
|  | *anomaly R* | 0.4 | 0.33 | 0.55 | 0.3 |
|  | RMSD | 15.21 | 19.33 | 10.45 | 18.71 |
|  | BIAS | -8.05 | 8.86 | -12.79 | 8.74 |
| ET (mm/mo) | *R* | 0.53 | 0.6 | 0.53 | 0.58 |
|  | *anomaly R* | 0.32 | 0.23 | 0.23 | 0.19 |
|  | RMSD | 20.89 | 23.47 | 18.45 | 24.56 |
|  | BIAS | 0.9 | 26.85 | 0.41 | 31.97 |
| NPP (g/m$^2$/mo) | *R* | 0.75 | 0.65 | 0.84 | 0.62 |
|  | *anomaly R* | 0.46 | 0.42 | 0.59 | 0.38 |
|  | RMSD | 36.78 | 35.9 | 19.53 | 34.32 |
|  | BIAS | 16.93 | 60.14 | -0.62 | 63.29 |
| GPP (g/m$^2$/mo) | *R* | 0.72 | 0.6 | 0.83 | 0.58 |
|  | *anomaly R* | 0.52 | 0.38 | 0.79 | 0.31 |
|  | RMSD | 48.01 | 47.63 | 25.42 | 45.95 |
|  | BIAS | 9.64 | 64.88 | -12.51 | 67.11 |