# Peer review of "Towards Effective Drought Monitoring in the Middle East and North Africa (MENA) Region: Implications from Assimilating Leaf Area Index and Soil Moisture into the Noah-MP Land Surface Model for Morocco"

_Hydrology and Earth System Sciences, 2021_

## Author Comment (AC1)

Response to reviewer #1

**General comment:**

Drought monitoring and early warning systems are valuable tools to enhance our understanding and better inform relevant authorities to act early and effectively to mitigate adverse effects that drought may bring to food security systems. Implementing strategies aimed at improving the reliability of Drought monitoring and early warning systems remains key. Authors of this manuscript have implemented an approach of assimilating remotely sensed data (LAI and soil moisture) to a land surface model (Noah-MP) to improve drought monitoring in the MENA region. The study is interesting and valuable not only for the MENA region but could also be applied in other regions experiencing drought challenges. Thus, the manuscript could contribute to valuable knowledge that this journal aims for. However, I have noted some scientific concerns (approaches and discussions) in the manuscript that could lower the quality of this manuscript at the current state.

**Response:** We thank the reviewer for the positive feedback to our study and please see our responses in more detail below.

**Specific scientific comment:**

1. Application of SMAP-DA must have failed due to assumptions taken in the model which were, highly likely, not representative of actual position on the ground. Model failure, especially from SSM-DA, was therefore almost certain. For this reason, the authors then focused on quantifying how data assimilation differentiates the categorization of drought and reproduces the evolution, duration, and intensity of past drought events as an indirect way to evaluate its impact on root zone soil moisture, which is good. However, the authors should first better highlight more on these assumptions, such as assuming soil is always irrigated to field capacity (lines 168 to 169) and uncertainty in irrigation information (irrigation frequency/timing and amount), as possible reasons behind DA failure. Could better representation of actual ground information in the model lead to improvements in model simulation after implementing SSM-DA? So in my view the authors should explicitly state that the reasons behind model failure was due to missing or limited in-situ data and local information and the decision to set the model with conditions that may not be the actual position on the ground.

**Response:** We thank the reviewer for the helpful suggestions. We now have added to our discussion regarding 1) uncertainties in the irrigation scheduling based on the soil moisture deficit approach in section 2.3 lines 180-182; and 2) possible reasons contributing to the failure of utilizing soil moisture data assimilation in improving

modeling performance for this case study in section 4 lines 532-534. We agree with the reviewer that the accessibility to in situ observational data with respect to irrigation scheduling and/or remote sensing soil moisture product at a resolution that can detect irrigation signal may all contribute to a better model configuration for the study region. Moreover, the information utilization in the current SSM-DA setup is suboptimal, which may also affect the efficiency of data assimilation (Nearing et al., 2018). For instance, the CDF-matching scaling approach prior to DA may have removed most of the signals. In areas where irrigation and agricultural practices dominate, significant revisions to SSM-DA strategy are required. We are working on a project investigating the potential of utilizing disaggregated SMAP products to detect irrigation scheduling and improving the modeling of irrigation, but this work is beyond the scope of the current study, and we look forward to applying the approach to this data limited region in the future, if positive results can be found for our more data intensive testbed region.

2. Authors should demonstrate or reference other studies on accuracy and uncertainty of the RS datasets, such as MODIS, used in this study.

**Response:** We have now revised both sections 2.2.1 and 2.2.2 by citing references that have evaluated the accuracy and uncertainties of the SMAP L3_E product and MODIS LAI product.

3. In line 149 to 152, the authors relied on irrigation maps from global datasets. The study area is relatively small. Why couldn't they generate more accurate irrigation maps? Did they attempt to validate the irrigation map from global dataset?

**Response:**  We acknowledge that there are uncertainties in irrigation mapping for our study region by relying on global datasets, which may be drawn from coarse report and satellite-based data (e.g., MODIS available at 500 m resolution), though GIA also incorporates sub-1km satellite information, and may not have been originally validated for this region. We spent much effort searching for possible local irrigation datasets for both model development and evaluation, including reaching out to our MENA partners, but we were unable to obtain such local maps at this time.

When developing our composited irrigation map from the GRIPC, GIA and GIAM datasets, we did check and verify the individual datasets and the composited map against Google Earth's imagery and available irrigation map plots published in the literature, including the following references:

Fig. 3.2 from Molle et al. (2019), which is based on irrigation information (circa 2010) from Morocco's MAPM (Ministère de l'Agriculture et de la Pêche Maritime). (2012). Place de l'eau dans le Plan Maroc Vert. Powerpoint

Fig. 1 from Kharrou et al. (2021) was later used for additional regional verification.

We have updated the text in the revised manuscript in lines 162-164 to now include the following statement:

"The final composite irrigation fraction map was verified against other imagery (such as Google Earth) and published Morocco irrigation maps (e.g., Figure 3.2 from Molle et al., 2019), and it is shown in Figure 1(b)."

4. In lines 306 to 310, the authors noted spatial variability of improvements in results obtained but failed to discuss the reasons behind these? Could they discuss this?

**Response:** We thank the reviewer for bringing up this question. The impact of data assimilation on total ET is largely determined by their relative impact on E and T component respectively as well as the ratio of E/T for the area. For instance, for grid cells that are classified as croplands, the magnitude of T is much larger than that of E. As LAI-DA leads to great improvement in T and T is the dominant component of ET for croplands, the overall performance for ET is improved. The dominance of transpiration in terrestrial ET especially for croplands are also documented in other studies (Jasechko et al., 2013; Kumar et al., 2020). We have added text in lines 321-323 to reflect this response.

5. Authors have stated in lines 351 to 352 that overestimation of transpiration during summertime, especially for croplands, likely due to misrepresentation of the vegetation seasonality. Why were there no attempts to better represent the vegetation seasonality in the model?

**Response:** There are indeed several efforts that have been made to address the weakness in the prognostic phenology module of Noah-MP from different perspectives (Li et al., 2021; Niu et al., 2020). In general, one factor might be contributing to this weakness is the oversimplification of parameterizing the effect of soil water stress on transpiration and carbon assimilation. Noah-MP, as well as many other earth system models, uses a β factor to control stomatal resistance and photosynthesis, which is a function of either soil moisture or matric potential depending on the options (Niu et al., 2011). However, this assumption has been shown to have resulted in large uncertainties (Trugman et al., 2018) and can systematically overestimate the effect of soil moisture drought on evaporative fluxes (Bonan, 1996). Another factor might be due to the limitation of static root profiles, which disconnects the interactions between changes in belowground water and nutrient resources and above ground plant carbon assimilation (Niu et al., 2020). The penalty of this assumption is even more obvious in dryland ecosystems or ecosystems during droughts as in reality plants under these conditions show much stronger resiliency to tolerate water stress (Ahlström et al.,

2015; Fensholt et al., 2012). For instance, using the default prognostic phenology module, Ma et al. (2017) reported that Noah-MP simulates lower-than-observer GPP, LAI and ET during droughts for Central America and Niu et al. (2020) reported that Noah-MP lacks the ability to simulate the plant resilience to stress under the Texas drought in 2011.

Realizing the importance of the role of vegetation in regulating the hydrological and carbon fluxes especially under stressed conditions, emerging studies started to improve the parameterization of soil-vegetation-atmospheric interactions in Earth System Models. For Noah-MP, for instance, efforts have been made such as introducing plant hydraulics scheme (Li et al., 2021), modifying the β factor (Niu et al., 2020), including crop modules and dynamic rooting depth (Liu et al., 2020; 2016). However, parameter calibration and retrieval for these implementations are heavily relied on in situ observational datasets, which is challenging for the MENA region, where there is limited or inaccessible observational data for supporting this research. Perhaps using satellite observations to aid the derivation of plant related parameters, such as plant hydraulic traits would be one of the future directions to improve the model representation for data scarce regions. The improvement of the modeling structure would in turn provide extended benefits to data assimilation. However, these are beyond the scope of the current study.

Besides the weakness in the prognostic phenology module, human management is another factor that may shift the seasonality of vegetation condition, which is often hard to capture with models alone, in such cases, integrating models with improved representation of both natural and human regulated processes and satellite observed measurements may all contribute to a better representation of the fluxes and states.

6. Authors have stated in line 431 that limited in-situ soil data was available. what were the sources of soil moisture data? Can they also include these data in the manuscript? what about using rainfall data as proxy for soil moisture, standardised precipitation index? did they attempt to use this?

**Response:** We apologize for any confusion. We meant that there are no in situ observational soil moisture measurements available for this domain. We thank the reviewer for bringing up the idea. We agree that SPI could be a proxy for soil moisture and could be a good candidate index to compare with in terms of characterizing drought. However, as the cross comparison for LAI-DA and SSM-DA resulted soil moisture drought indicator for this study is only based on the period of 2002-2019, which might be too short of a period as SPI is normally computed based on a long-term data record. Besides, we have tried to request data from MENA partners and local stakeholders regarding evaluating the drought characterization, but such data were not

provided. So, rather than finding and relying on a qualified reference to evaluate the data assimilation impact on drought characterization, here the goal is more towards comparing the difference in characterizing soil moisture-based drought between LAI-DA and SSM-DA. The evaluation for the data assimilation performance is more focused on ET, GPP, and NPP, which are fluxes that can be evaluated with relatively reliable remote sensing reference datasets.

7. Why are all figures placed at the end of the manuscript rather than at their rightful positions? This approach makes it difficult for the reader to follow smoothly.

**Response:** We apologize for the inconvenience it has brought for the review process. We will submit our revised version with figures and tables inserted in the main text near the location of the first mention.

8. Line 411, authors are referring to the figure 2(b), representing the difference of correlation (R) – computed as DA minus OL for the period of 2015-2019 for LAI-DA in terms of transpiration? why specifically stratify this figure?

**Response:** We apologize for the typo. It should be figure 1 (b), which indicates the irrigation fraction intensity and the distribution of the lightly, moderately, and heavily irrigated area. We now corrected this typo in the revised manuscript.

**Technical corrections:**

1. Line 65, separate DA and reference using separate brackets.

**Response:** Modified.

2. Line 191, state SSM-DA and SSM-DA$_{irr}$ separately and not as SSM-DA (SSM-DA$_{irr}$). line 197 it is well stated LAI-DA and LAI-DA$_{irr}$.

**Response:** Modified.

3. Line 214, correct the superscript of degree in 0.05o

**Response:** Modified.

4. Line 256, SIF is not defined but then defined later in line 260. it should be defined at this point

**Response:** Modified. SIF is now defined in the first paragraph of section 2.5 when it appears for the first time.

5. Line 392, place irr as a subscript in LAI-DA$_{irr}$

**Response:** Modified.

**References**

Ahlström, A., Raupach, M. R., Schurgers, G., Smith, B., Arneth, A., Jung, M., Reichstein, M., Canadell, J. G., Friedlingstein, P. and Jain, A. K.: The dominant role of semi-arid ecosystems in the trend and variability of the land CO2 sink, Science, 348(6237), 895–899, 2015.

Bonan, G. B.: Land surface model (LSM version 1.0) for ecological, hydrological, and atmospheric studies: Technical description and users guide. Technical note,, 1996.

Fensholt, R., Langanke, T., Rasmussen, K., Reenberg, A., Prince, S. D., Tucker, C., Scholes, R. J., Le, Q. B., Bondeau, A. and Eastman, R.: Greenness in semi-arid areas across the globe 1981–2007—an Earth Observing Satellite based analysis of trends and drivers, Remote Sensing of Environment, 121, 144–158, 2012.

Jasechko, S., Sharp, Z. D., Gibson, J. J., Birks, S. J., Yi, Y. and Fawcett, P. J.: Terrestrial water fluxes dominated by transpiration, Nature, 496(7445), 347–350, 2013.

Kharrou, M. H., Simonneaux, V., Er-Raki, S., Le Page, M., Khabba, S. and Chehbouni, A.: Assessing Irrigation Water Use with Remote Sensing-Based Soil Water Balance at an Irrigation Scheme Level in a Semi-Arid Region of Morocco, Remote Sensing, 13(6), 1133, 2021.

Kumar, S. V., Holmes, T. R., Bindlish, R., Jeu, R. de and Peters-Lidard, C.: Assimilation of vegetation optical depth retrievals from passive microwave radiometry, Hydrology and Earth System Sciences Discussions, 24(7), 3431–3450, 2020.

Li, L., Yang, Z. L., Matheny, A. M., Zheng, H., Swenson, S. C., Lawrence, D. M., Barlage, M., Yan, B., McDowell, N. G. and Leung, L. R.: Representation of Plant Hydraulics in the Noah-MP Land Surface Model: Model Development and Multi-scale Evaluation, J. Adv. Model. Earth Syst., 1–57, doi:10.1029/2020MS002214, 2021.

Liu, X., Chen, F., Barlage, M. and Niyogi, D.: Implementing Dynamic Rooting Depth for Improved Simulation of Soil Moisture and Land Surface Feedbacks in Noah-MP-Crop, J. Adv. Model. Earth Syst., 12(2), 1–15, doi:10.1029/2019MS001786, 2020.

Liu, X., Chen, F., Barlage, M., Zhou, G. and Niyogi, D.: Noah-MP-Crop: Introducing dynamic crop growth in the Noah-MP land surface model, J. Geophys. Res. Atmos., 121(23), 13,953–13,972, doi:10.1002/2016JD025597, 2016.

Ma, N., Niu, G. Y., Xia, Y., Cai, X., Zhang, Y., Ma, Y. and Fang, Y.: A systematic evaluation of Noah-MP in simulating land-atmosphere energy, water, and carbon exchanges over the continental United States, J. Geophys. Res. Atmos., 122(22), 12–245–12–268, 2017.

Molle, F. and Sanchis-Ibor, C.: Irrigation policies in the Mediterranean: Trends and challenges,, 279–313, 2019.

Nearing, G., Yatheendradas, S., Crow, W., Zhan, X., Liu, J. and Chen, F.: The efficiency of data assimilation, Water Resour. Res., 54(9), 6374–6392, 2018.

Niu, G. Y., Fang, Y. H., Chang, L. L., Jin, J., Yuan, H. and Zeng, X.: Enhancing the Noah-MP Ecosystem Response to Droughts with an Explicit Representation of Plant Water Storage Supplied by Dynamic Root Water Uptake, J. Adv. Model. Earth Syst., 1–29, doi:10.1029/2020MS002062, 2020.

Niu, G. Y., Yang, Z. L., Mitchell, K. E., Chen, F., Ek, M. B., Barlage, M., Kumar, A., Manning, K., Niyogi, D., Rosero, E., Tewari, M. and Xia, Y.: The community Noah land surface model with multiparameterization options (Noah-MP): 1. Model description and evaluation with local-scale measurements, J. Geophys. Res., 116(D12), 1381–19, doi:10.1029/2010JD015139, 2011.

Trugman, A. T., Medvigy, D., Mankin, J. S. and Anderegg, W.: Soil moisture stress as a major driver of carbon cycle uncertainty, Geophys. Res. Lett., 45(13), 6495–6503, 2018.

---

## Author Comment (AC2)

Response to reviewer #2

**General Comments:**

This manuscript analyses the impact of assimilating LAI and soil moisture data from remote sensing (separately) on the water and carbon fluxes simulated by a land surface model (Noah-MP) and how drought categorization is affected as a result. The impact of irrigated areas is also included in the analysis through a simple irrigation module. This is a very relevant topic as early drought warning can help implement mitigation measures to reduce adverse impacts. Irrigation is often not included in LSMs, and its inclusion here is valuable in bringing focus to the difficulties encountered in its inclusion in models, in particular in data scarce regions.

The work is well written and the evidence is clearly and convincingly presented.

**Response:** We thank the reviewer for the encouraging comments on our study and please see our responses in detail below.

**Specific Comments/Questions:**

1. There is no description of the downscaling of the SMAP/upscaling of the MODIS LAI data. A discussion on the spatial resolution of the datasets is also missing: what could be its impact relative to 1. the model grid and 2. the landscape fragmentation (in particular size of irrigated areas)?

**Response:** In this study, we use bilinear interpolation for spatial disaggregation and averaging for upscaling methods to regrid SMAP L3_E soil moisture product at 9 km and the MODIS LAI datasets at 500 m to 0.05° in model space, respectively. We agree that the differences in spatial resolution in the input and assimilated datasets may affect the model performance and its capability to capture fine scale variabilities. For instance, some of the irrigated areas are sparsely distributed and may not be reasonably represented by a 0.05° grid. The model resolution could go finer to 1 km along with the support of the input datasets at comparable spatial scales, however, that might not be necessary for SSM-DA as the resolution for the SMAP L3_E product is only 9 km. Therefore, we think the choice of 5 km is reasonable to capture major signals for water-energy-carbon cycle in response to data assimilation and irrigation. The sensitivity studies regarding the spatial resolution at which DA is performed could be an interesting point to explore but is beyond the scope of this study. We now added this information of downscaling/upscaling of the assimilated datasets in line 117 and lines 137-138.

2. The parameterization of the irrigated areas and module deserves more discussion: what could be the impact of the uncertainty of the global datasets used? How about irrigation amounts which assume reaching of field capacity? I also do not see a mention of matching the irrigated areas to the land use map, were irrigated areas only applied to pixels identified as cropped/partially cropped?

**Response:** These are great questions, and we address each one in turn.

1) what could be the impact of the uncertainty of the global datasets used?

The uncertainty coming from global irrigation maps can impact spatially where irrigation may be realistically located and in relation to uncertainties from methods in deriving the landcover types, since many of the datasets are derived independently from each other and sometimes with different training datasets. We accept that uncertainty is inherent in all modeling studies, even at fine resolutions given local observational data, and that is why we had compared and combined the most common spatial attributes found in the irrigation and landcover maps, to reduce such uncertainties. We also addressed a similar question from Reviewer 1 -- question #3, which we hope helps further address your question here.

2) How about irrigation amounts which assume reaching of field capacity?

The assumption of bringing irrigation to field capacity has been evaluated for CONUS, which has yielded good results, but it may inherit some uncertainties when being applied to Morocco. We do lack in situ data to optimize our irrigation parameter settings. Also, utilizing other datasets, such as a finer scale soil moisture product to inform irrigation timing and targeted soil moisture conditions, would be helpful.

3) I also do not see a mention of matching the irrigated areas to the land use map, were irrigated areas only applied to pixels identified as cropped/partially cropped?

To address your final question, we want to mention that we do match irrigated areas with certain landcover types from the land use map, mainly the cropland and grassland classes. This comparison has an initial check implemented when deriving the composited irrigation map and processed through LDT, and then again when the irrigation is applied in LIS, following the method outlined in Ozdogan et al. (2010). We now updated the text in the revised manuscript in lines 164-166 with the following statement: "Note that to avoid potential mismatch between the land cover type and irrigation fraction, an initial check has been implemented in both LDT and LIS to constrain irrigation within certain land cover types mainly the cropland and grassland classes."

3. There is a brief mention of loss of information due to the rescaling of the soil moisture product to the model climatology but I think this deserves further discussion. What is the impact of rescaling the SSM to a model run which does not include irrigation? In particular when there is also an attempt to use the SSM to improve the irrigation run. A comparison of the SMAP cdf over an irrigated vs. non-irrigated pixel would be an interesting starting point.

**Response:** We thank the reviewer for bringing up this question. We acknowledge that scaling soil moisture product to the model climatology in absence of irrigation representation may lose information especially for places where SMAP can detect irrigation signal. These issues may all limit the efficiency of data assimilation and have been acknowledged and discussed in prior studies (Kumar et al., 2015; Nearing et al., 2018). In our SSM-DA and SSM-DA$_{irr}$ experiments, the SMAP L3_E product is scaled to the climatology of OL and OL$_{irr}$ respectively to provide an overview of the model performance in scaling and assimilating soil moisture with/without irrigation representation. We also notice that the situation could be the opposite – that irrigation is represented in the model but was not detectable by SMAP. And it is challenging to come to a conclusion as both the irrigation parameterization in the model and the SMAP product are subject to errors in terms of capturing irrigation timing and frequency. Lacking of in situ observations for model calibration poses further challenges to improve the irrigation parameterization and the nature of sparsely located irrigation farms for the study region limits the benefit of SMAP in detecting irrigation signals for Morocco due to its relatively coarse spatial resolution. We now populate the discussion by listing possible reasons contributing to the failure of utilizing soil moisture data assimilation in improving modeling performance for this case study in section 4 lines 532-534.

4. Was a simultaneous assimilation of LAI and SSM tested? If not, why not? Do you see any potential for the inclusion of both to improve the results?

**Response:** We thank the reviewer for bringing up this question. In fact, joint SSM and LAI assimilation is initially planned for this study and we've tried various experiments attempting to find values from digesting the SMAP soil moisture information but unfortunately, we didn't see any encouraging messages from joint DA for the Morocco domain likely because the information utilization in the current SSM-DA setup is suboptimal and in areas such as this where irrigation and agricultural practices

dominate, significant revisions to the SSM-DA strategy is required. But we do agree that potentially there are regions that can benefit from joint SSM and LAI DA.

**Technical Comments**

l.268: 'for the' should be removed (or rephrase sentence, unclear)

**Response:** The sentence is now rephrased as "Statistical skill metrics include the Pearson's correlation (R) and anomaly correlation (anomaly R) coefficients based on monthly time series with 95% significance tested using Fisher's z transform test..."

l.300: 'with the percent change over 20%' is awkward. Suggestion: 'with a relative improvement of over 20%'

**Response:** Modified.

458-460: I assume the higher LAI leads to increased transpiration. The sentence structure implies the opposite effect. Add 'for LAI-DA' for clarity at the end of the sentence.

**Response:** Modified.

**References**

Kumar, S. V., Peters-Lidard, C. D., Santanello, J. A., Reichle, R. H., Draper, C. S., Koster, R. D., Nearing, G. and Jasinski, M. F.: Evaluating the utility of satellite soil moisture retrievals over irrigated areas and the ability of land data assimilation methods to correct for unmodeled processes, Hydrol. Earth Syst. Sci., 19(11), 4463–4478, doi:10.5194/hess-19-4463-2015, 2015.

Nearing, G., Yatheendradas, S., Crow, W., Zhan, X., Liu, J. and Chen, F.: The efficiency of data assimilation, Water Resour. Res., 54(9), 6374–6392, 2018.

---

## Author Response (AR1)

Response to Editor:

Due diligence was given to address the review comments provided by the reviewers. However, some basic facts need to be corrected before taking a decision.
First of all the authors need to reconsider reworking on the whole study by incorporating the actual resolution of the SMAP L3 soil moisture product, that is ~33 km. Considering this aspect, the actual resolution of~33 km may impact the overall results/outcome of the study.

**Response:** We thank the editor for this comment and we agree that the sampling resolution of the SMAP data (~33km) may not be sufficient to capture the irrigation features in many places. Prior studies such as Lawston et al. [2017] have documented that the SMAP L3 enhanced product is able to represent large scale irrigation features, which is the primary reason for its use in this study. In our own study domain, the SMAP L3 enhanced product shows increased utility compared to the use of the standard SMAP L3 product. We did not include these results in the manuscript as the structure of the experiments using the standard and enhanced product is not exactly equivalent (the forcing datasets, model spinup strategies, etc.).

The authors have admitted that having high-resolution information will improve effective monitoring of drought over the North Africa region. The SMAP mission also provides the SMAP-Sentinel based soil moisture product at 3 km and 1 km with a revisit interval between 6-12 days. Evaluation of irrigation signals from this high-resolution SMAP product is also crucial because the spatial resolution is important when it comes to the detection of irrigation signals and their influence on drought detection over croplands.

**Response:** We thank the editor for providing this valuable information and we agree that assimilating high-resolution SMAP-Sentinel product has the potential to improve the soil moisture condition impacted by irrigation, thus providing better surface condition for Morocco, and benefiting the drought monitoring. However, the assimilation of SMAP-Sentinel products is not supported in the NASA LIS framework, which is a non-trivial effort. However, we acknowledge that this is an important topic for soil moisture data assimilation, and we are interested in applying the product to our other on-going projects as we work towards implementing the feature of assimilating SMAP-Sentinel products in our modeling framework. To highlight this potential, we elaborated our discussion by adding the following sentences in section 4 para 3:

"Although SMAP products are reported to show capability in detecting irrigation signal for places such as California Central Valley and High Plains [Lawston et al., 2017;

Felfelani et al., 2018; Kumar et al., 2018], this capability is likely to be limited within the intensively irrigated hot spots that have limited spatial extents. To capture the irrigation signal for smaller or sparsely distributed irrigation areas, soil moisture products at higher resolution has greater potential to provide benefits such as the SMAP-Sentinel1 datasets [Lievens et al., 2017; Das et al., 2019; Jalilvand et al., 2021].

**References:**

Das, N. N., D. Entekhabi, R. S. Dunbar, M. J. Chaubell, A. Colliander, S. Yueh, T. Jagdhuber, F. Chen, W. Crow, and P. E. O'Neill (2019), The SMAP and Copernicus Sentinel 1A/B microwave active-passive high resolution surface soil moisture product, Remote Sensing of Environment, 233, 111380.

Felfelani, F., Y. Pokhrel, K. Guan, and D. M. Lawrence (2018), Utilizing SMAP soil moisture data to constrain irrigation in the Community Land Model, Geophys. Res. Lett., 45(23), 12–892–12–902.

Jalilvand, E., R. Abolafia-Rosenzweig, M. Tajrishy, and N. N. Das (2021), Evaluation of SMAP/Sentinel 1 High-resolution soil moisture data to detect irrigation over agricultural domain, IEEE Journal of Selected Topics in Applied Earth Observations and Remote Sensing, 14, 10733–10747.

Kumar, S. V., P. A. Dirmeyer, C. D. Peters-Lidard, R. Bindlish, and J. Bolten (2018), Information theoretic evaluation of satellite soil moisture retrievals, Remote Sensing of Environment, 204, 392–400.

Lawston, P. M., J. A. Santanello Jr, and S. V. Kumar (2017), Irrigation signals detected from SMAP soil moisture retrievals, Geophys. Res. Lett., 44(23), 11–860–11–867.

Lievens, H., R. H. Reichle, Q. Liu, G. J. M. De Lannoy, R. S. Dunbar, S. B. Kim, N. N. Das, M. Cosh, J. P. Walker, and W. Wagner (2017), Joint Sentinel-1 and SMAP data assimilation to improve soil moisture estimates, Geophys. Res. Lett., 44(12), 6145–6153, doi:10.1002/2017GL073904.

---

## Author Response (AR2)

Response to reviewer #3

**General comment:**

In this study, SMAP 9km soil moisture and MODIS LAI are assimilated with the Noah-MP land surface model, and the improvement in the simulation of the ET and carbon fluxes alongside the root zone soil moisture-based drought index is examined over the MENA region. As irrigation contributes significantly to the seasonal ET and carbon flux, the model performance in simulating the parameter mentioned above after activating the irrigation module is also investigated. The paper is interesting, thorough, and well written. In my opinion, the authors have addressed the objectives of the study. However, there are a few concerns regarding the choice of the satellite soil moisture product for assimilation and the evaluation datasets that are listed below. Addressing these concerns, I recommend acceptance.

**Response:** We thank the reviewer for the positive feedback to our study. Please see our responses in more detail below.

**Major comments:**

1- L110) Most of the agricultural areas around the world are smaller than 0.5 km2; thus, it is possible that the coarse resolution soil moisture product such as SMAP enhanced 9km can not capture the irrigation signal. Thus, assimilating the soil moisture data might not improve the irrigation simulation, leading to underestimation of ET over cropland area. My question is why the authors did not consider using a higher resolution soil moisture product such as SMAP - Sentinel1 1km soil moisture product that is proven to contain the irrigation signal over the irrigated cropland area (Jalilvand et al., 2021). I see that in conclusion, it is mentioned that SMAP-Sentinel1 1km soil moisture data can be used in the future study, but a justification for why you have chosen to work with the 9km product in this study is needed here.

**Response:** We thank the reviewer for this comment. Selecting the SMAP product for this research is mainly inspired by the following aspects: 1) the product has been proved to have higher information content compared to other products such as ASCAT and SMOS [*Kumar et al.,* 2018], 2) it has capability of detecting irrigation signal especially for semi-arid regions with Mediterranean climate [*Lawston et al.,* 2017], and 3) it has the capability to constrain and improve irrigation simulation in LSMs [*Felfelani et al.,* 2018]. A justification statement is added in Lines 115-116 as follows:

*"The product has also been proved to have higher information content with capability in detecting irrigation signals and improving irrigation simulations in large-scale LSMs (Kumar et al., 2018; Lawston et al., 2017; Felfelani et al., 2018)."*

We agree with the reviewer that utilizing soil moisture product at a higher spatial resolution such as SMAP-Sentinel1 datasets may provide more benefits in capturing the irrigation signal and its impact on surface conditions and we look forward to incorporating and testing these products within LIS in future projects.

2- L250: In contrast with what is mentioned here, a study by Javadian et al., 2019, over a basin in Iran (which is located in the MENA region), showed that WaPOR is systematically underestimated the ET over the irrigated cropland area while having more accurate estimates over rainfed agriculture and bare soil. This can significantly impact the results reported in this study as the WaPOR is used as the evaluation dataset for E, T, ET, and NPP. Please comment on this.

**Response:** We thank the reviewer for providing this information. Generally, it is safe to assume that there is no state-of-art ET product that can accurately represent the ET magnitudes. Therefore, our analyses for ET and its components are more focused on evaluating their seasonal to interannual variabilities. We acknowledge that the comparison associated with the magnitude of ET such as RMSE and BIAS should be interpreted carefully along with evidence of consistency on other variables such as GPP. The comparison on ET magnitude is discussed by acknowledging the uncertainties from both WaPOR dataset and the model output, which are noted in Line 327-329, and 371-373 as follows:

*Lines 327-329: "We note that both FAO WaPOR and FLUXSAT data sets are remote-sensing model data-driven products and are thus subject to uncertainties. However, the consistent results obtained by comparing the carbon fluxes against the two independent data sources highlight the benefit of assimilating LAI into the system."*

*Lines 371-373: "This may stem from different ET partitioning algorithm between Noah-MP and WaPOR as well as their associated uncertainties. The different impact on E and T due to LAI assimilation results in overall small difference in terms of correlation while general improvements in terms of RMSD for ET."*

Noah-MP OL simulation show overall positive BIAS in ET during the growing seasons across all three land cover types. LAI assimilation reduces the magnitude of BIAS, but the sign is still positive. WaPOR is regarded as one reasonable reference dataset rather than an approximation to the ground truth. A thorough investigation on the ET magnitude accuracy could not be achieved without integrating multi-source measurements, which is limited for the region and is beyond the scope of our study.

3- Figure 3b) Following the first comment, one important difference between the LAI and SMAP datasets is that the native MODIS LAI resolution is 500 m, upscaled to 0.05 degrees, while the SMAP enhanced product native resolution is coarser than 9 km resampled to the 0.05 degree.

Thus, the LAI data may contain the subpixel information while SMAP does not. This is especially important over cropland areas where land cover and soil moisture constantly change due to a different land and water management at the plot scale. This might be another explanation for the greater impact of LAI-DA on improving the T simulation relative to the SSM-DA. Please comment on this.

**Response:** We thank the reviewer for this comment. We agree that the spatial resolution of SMAP is one of the important factors that limits the improvement in transpiration simulation. Besides, the representation of soil moisture and ET coupling is known to have issues in land surface models that an improved soil moisture condition may not be effectively converted to improved ET [*Crow et al.,* 2020] and the contribution of assimilating soil moisture on improving ET may also depend on region and climate [*Kumar et al.,* 2020]. We now elaborated this discussion in Section 3.1.2, lines 344-349 as follows:

*"The impact of SSM-DA on ET components is limited. On the one hand, this may be resulted from the coarse spatial resolution of the SMAP dataset as it cannot provide information for finer scale soil moisture variability. On the other hand, it could also be possible that improved soil moisture condition in SSM-DA is not effectively converted to improved ET because of the weakness in model representation of ET and soil moisture coupling. This is an known issue for many land surface models (Crow et al., 2020). Moreover, the impact of soil moisture assimilation on ET can also heavily depend on region and climate (Kumar et al., 2020)."*

4- L368) According to Figure 5, LAI-DA significantly increases the RMSD relative to the OL run. If I understand correctly, you attribute this to using different ET algorithm. The only difference between the LAI-DA and OL run is the assimilation of LAI and not the difference in the ET algorithm. Can you explain why assimilation of LAI resulted in much larger RMSD?

**Response:** We apologize for the ambiguity here. We meant to say that Noah-MP and WaPOR have different ET algorithms, resulting in different partitioning of E and T. Assimilating LAI affects the greenness fraction in Noah-MP, which is one of the most sensitive parameters that controls the partition of E and T. Assimilating LAI reduces the LAI magnitude as compared to OL for all three land cover types (Figure 4), leading to increased fractional area for bare soil as well as increased evaporation. This is the main reason for increased RMSD for E in LAI-DA shown in Figure 5. To avoid ambiguity, we now rephrase the sentence in Line 371-373 in the revised manuscript as follows:

*"This may stem from different ET partitioning algorithm between Noah-MP and WaPOR as well as their associated uncertainties. The different impact on E and T due to LAI assimilation*

*results in overall small difference in terms of correlation while general improvements in terms of RMSD.*"

**Moderate comments:**

1. L195) Please summarize all the experiments information in one table. As I understand, you have 6 different experiments that can be shown in this table.

**Response:** We thank the reviewer for the suggestion. A table summarizing the simulations is now added to the revised manuscript.

2. L260: Is there any flux tower site in northern Morocco? If not, please explain why the FLUXCOM data is reliable over the study region.

**Response:** We thank the reviewer for bringing up this point. Unfortunately, FLUXCOM does not have sites over northern Morocco. This reference dataset is included in our analyses, along with FLUXSAT GPP and GOME SIF to provide multiple views regarding how the simulated carbon flux compared to available remote sensing data sets. As shown in Figure 6, the comparison with FLUXCOM GPP yields similar results as compared to FLUXSAT and GOME SIF data sets. We now modify Lines 263-264 in section 2.5.2 to note this issue:

*"As there is no flux tower site over the study domain and FLUXCOM products do not cover the SMAP period, we also utilize the recently developed FLUXSAT GPP estimates, which are available from 2000-2020."*

3. Figure 5) add the median to either y-axis label or the figure caption, so the readers know these are the median of the metrics shown for each month, as explained in L361.

**Response:** Caption of Figure 5 is modified as suggested.

4. Figure 5) I like to see the same plot for ET to see how the good and bad performance of LAI-DA in simulating T and E respectively cancel out each other on a monthly basis. Can you add this plot to the supplementary material?

**Response:** We thank the reviewer for the suggestion. The figure below includes the comparison for ET. In general, LAI-DA impact on the ET correlations are small while it leads to overall reduced RMSD especially during the growing seasons. We now update Figure 5 in the main text by including the ET correlation and RMSD and report the findings in Lines 372-373 as follows:

*"The different impact on E and T due to LAI assimilation results in overall small difference in terms of correlation while general improvements in terms of RMSD for ET."*

[Figure]

5. L364) Transpiration RMSD during the summertime and over cropland area is high. Can you explain why LAI-DA does not significantly improve the vegetation seasonality?

**Response:** We thank the reviewer for the comments. LAI-DA did improve the seasonality (Figure 3b) and interannual variability (Figure 5b) for transpiration. The high RMSD for croplands is more associated with the magnitude difference between Noah-MP and WaPOR, which may stem from different E/T partitioning algorithm between the two, different forcing inputs (Noah-MP driven by GDAS+IMERG vs. WaPOR driven by MERRA), and other associated uncertainties respectively.

6. L466) Based on figure 8, almost the same pattern is observed using OL run (stronger drought over cropland region and weaker over other landcover types), so I do not think this is an artifact of assimilating LAI.

**Response:** We agree that this pattern is not clearly shown in Figure 8. Here we provide the following figure showing the distribution of percentage area under >D3 drought during the 2018-2019 event for each land cover type. Here it can be seen that with LAI-

DA, the percent area under drought for croplands occupies a greater amount than in OL or SSM-DA.

[Figure]

7. L479) Based on Figure 9 it seems that the area under extreme drought (D3) is almost similar for both LAI-DA and SSM-DA over all the land cover types and the difference is limited to the exceptional drought category (D4).

**Response:** We thank the reviewer for pointing this out. We agree that this opposite tendency is more shown for D4 category, and we have revised the text accordingly in Lines 482-485 as follows:

*"More differences are seen in categorizing the moderate to extreme drought events (D1-D3) and there is no clear pattern associating with the differences. When it comes to the exceptional drought (D4), LAI-DA and SSM-DA show the opposite tendency as compared to OL for open shrublands and grasslands, in that LAI-DA tends to limit the spatial extent of the extreme and exceptional drought while SSM-DA is more likely to expand the impact of higher level of drought extremes."*

References:

Javadian, M.; Behrangi, A.; Gholizadeh, M.; Tajrishy, M. METRIC and WaPOR Estimates of Evapotranspiration over the Lake Urmia Basin: Comparative Analysis and Composite Assessment. Water 2019, 11, 1647. https://doi.org/10.3390/w11081647

E. Jalilvand, R. Abolafia-Rosenzweig, M. Tajrishy and N. N. Das, "Evaluation of SMAP/Sentinel 1 High-Resolution Soil Moisture Data to Detect Irrigation Over Agricultural Domain," in IEEE Journal of Selected Topics in Applied Earth Observations and Remote Sensing, vol. 14, pp. 10733-10747, 2021, doi: 10.1109/JSTARS.2021.3119228.

**References:**

Crow, W. T., C. A. Gomez, J. M. Sabater, T. Holmes, C. R. Hain, F. Lei, J. Dong, J. G. Alfieri, and M. C. Anderson (2020), Soil Moisture–Evapotranspiration Overcoupling and L-Band Brightness Temperature Assimilation: Sources and Forecast Implications, *Journal of Hydrometeorology*, *21*(10), 2359–2374, doi:10.1175/JHM-D-20-0088.1.

Felfelani, F., Y. Pokhrel, K. Guan, and D. M. Lawrence (2018), Utilizing SMAP soil moisture data to constrain irrigation in the Community Land Model, *Geophys. Res. Lett.*, *45*(23), 12–892–12–902.

Kumar, S. V., P. A. Dirmeyer, C. D. Peters-Lidard, R. Bindlish, and J. Bolten (2018), Information theoretic evaluation of satellite soil moisture retrievals, *Remote Sensing of Environment*, *204*, 392–400.

Kumar, S. V., T. R. Holmes, R. Bindlish, R. de Jeu, and C. Peters-Lidard (2020), Assimilation of vegetation optical depth retrievals from passive microwave radiometry, *Hydrology and Earth System Sciences Discussions*, *24*(7), 3431–3450.

Lawston, P. M., J. A. Santanello Jr, and S. V. Kumar (2017), Irrigation signals detected from SMAP soil moisture retrievals, *Geophys. Res. Lett.*, *44*(23), 11–860–11–867.